

# Past and future climate change effects on thermal regime and oxygen solubility of four peri-alpine lakes

Authors: Olivia Desgué-Itier[1], Laura Melo Vieira Soares[1], Orlane Anneville[1], Damien Bouffard[2], Vincent Chanudet[3], Pierre Alain Danis[4,5], Isabelle Domaizon[1], Jean Guillard[1], Théo Mazure[1], Najwa Sharaf[4,5], Frédéric Soulignac[6], Viet Tran-Khac[1], Brigitte Vinçon Leite[7], Jean-Philippe Jenny[1]

[1]Université Savoie Mont Blanc, INRAE, CARRTEL, 74200 Thonon-les-Bains, France
[2]Eawag, Swiss Federal Institute of Aquatic Science and Technology, Surface Waters – Research and Management, 6047 Kastanienbaum, Switzerland
[3]EDF, Hydro Engineering Centre, Environment and Social Department, 73290 La Motte Servolex, France
[4]Pôle R&D « Ecla », INRAE, 3275 Route Cézanne, 13182 Aix-en-Provence, France[5]Office Français de la Biodiversité, Unité « Ecla », INRAE, Aix-en-Provence, France
[5]INRAE, Aix Marseille Univ, RECOVER, Team FRESHCO, 3275 Route Cézanne, 13182 Aix-en-Provence, France
[6]CIPEL, International Commission for the protection of the waters of Lake Geneva, 1260 Nyon, Switzerland
[7]Laboratoire Eau, Environnement, Systèmes Urbains (LEESU), École Nationale des Ponts et Chaussées, Marne-la-Vallée, France

*Correspondence to*: Olivia Desgué-Itier (olivia.desgue@inrae.fr); Jean-Philippe Jenny (Jean-Philippe.Jenny@inrae.fr)

**Abstract.** Climate change modifies the thermal regime and the oxygen solubility of lakes globally, resulting in the alteration of ecosystem processes, lake habitats and concentrations of key parameters. The use of one-dimensional (1D) lake model for global scale studies has become the standard in lake research to evaluate the effects of climate change. However, such approach requires global scale forcing parameters which have several limitations that are barely discussed, such as the need of serious downscaling. Furthermore, projections of lakes' thermal regime are hardly ever confronted with long-term observations that extent for more than a few decades. These shortfalls limit the robustness of hindcast/ forecast simulations on decadal to centennial timescales. In this study, several 1D lake models' robustness was tested for long-term variations based on 63 years of limnological data collected by the French Observatory of LAkes (OLA). Here we evaluate the possibility to force mechanistic models by following the long-term evolution of shortwave radiation and air temperature while providing realistic seasonal trend for the other parameters for which local scale downscaling often lacks accuracy. Then, the effects of climate change on the thermal regime and oxygen solubility were analyzed in the four-largest French peri-Alpine lakes. Our results show that 1D lake models forced by air temperatures and short-wave radiations accurately predict variations in lake thermal regime over the last four to six decades, with RMSE <1.95 °C. During the last three decades, water temperatures have increased by 0.46 °C decade$^{-1}$ (±0.02 °C) in the epilimnion and 0.33 °C decade$^{-1}$ (±0.06 °C) in the hypolimnion. Concomitantly and due to thermal change, $O_2$ solubility has decreased by -0.104 mg L$^{-1}$ decade$^{-1}$ (±0.005 mg L$^{-1}$) and -0.096 mg L$^{-1}$ decade$^{-1}$ (±0.011 mg L$^{-1}$) in the epilimnion and hypolimnion, respectively. Based on the ssp370 socio-economic pathway of the IPCC, perialpine lakes could face an increase of 3.80 °C (±0.20 °C) in the next 70 years, accompanied by a decline of 1.0 mg L$^{-1}$ (±0.1 mg L$^-$



[1]) of $O_2$ solubility. These results suggest important degradation in lake thermal and oxygen conditions and a loss of habitats for endemic species.

## 1 Introduction

Lakes are critical ressources providing humanity with many ecosystem services such as hydropower and drinking water production (Jenny et al., 2020) and are considered sentinels of climate change (Williamson et al., 2009). Nevertheless, the alteration of these ecosystems under anthropogenic pressures and ongoing global warming requires continuous water quality monitoring. Lake water temperature is a critical indicator for long-term monitoring of lake ecosystems and for adapting better management practices for preservation (Daufresne et al., 2009). It is an important parameter impacting the metabolism, composition, and functioning of lake ecosystems as well as reflecting their response to climate change. Water temperature has a direct effect on the speed of development, growth rate and reproduction rate of aquatic organisms (Angellita et al., 2003 ; Mari et al., 2016a) and on phenology (Parmesan, 2006 ; Walter et al., 2002). Furthermore, water temperature can also indirectly affect organisms, by altering the mixing regime of lakes and governing gas solubility with potential impacts on oxygenation conditions, one of the most fundamental parameters of life in lakes (Roberts et al., 2009a; Wetzel, 2001a).

Recent global studies from lakes around the world have already shown that increasing air temperature has a significant effect on the intensification and the duration of stratification (Woolway and Merchant, 2019; Piccioni et al., 2021; Woolway et al., 2021). In deep temperate lakes, vertical mixing of the water column has experienced a decrease in intensity, frequency, and duration (Råman Vinnå et al., 2021; Danis et al., 2004), thereby increasing the vertical temperature gradient between the surface and deep layers (Livingstone, 2003). However, the study of lakes' thermal regime over decadal to centennial timescales is still limited, hence precluding our understanding of the evolution of lake physicochemical conditions and habitats as well as the response mechanisms to the forcings involved over such timescales.

Mechanistic lake models have been widely implemented over the last years (Bruce et al., 2018; Shatwell et al., 2019; Snortheim et al., 2017) and are considered as an essential tool for understanding, analyzing, testing different scenarios, and predicting the state of an ecosystem under external constraints over different timescales (Trolle et al., 2012). More specifically, the use of one-dimensional (1D) model for global scale studies has become the standard in lake research. These models are suitable to simulate complex ecosystems as they require minimum configuration parameters (Hamilton and Schladow, 1997; Vinçon-Leite et al., 2014), and to perform reliable predictions to study lakes responses to global warming (Balsamo et al., 2012). However, such approach requires global scale forcing parameters which has several limitations that are barely discussed. Among these limitations, the influence of the rivers and watersheds is not systematically assessed whereas it can have a strong effect on the change in the thermal structure of lakes with a short residence time (e.g <1-3 year) (Råman Vinnå et al., 2017, 2018; Perga et al., 2018). Another limitation relies on the quality of the forcing. Lakes are often smaller than the meteorological grid size used in studies. Some parameters such as wind can strongly vary at local scales especially in alpine regions. This issue is attimes tackled by using meteorological forcing downscaled to local weather stations (e.g. (Råman Vinnå





et al., 2021), or by trying to simplify the heat budget (e.g. (Piccolroaz et al., 2013). The quality of the input data is also limiting the possibility to estimate changes in the thermal structure over very long trends (> 100 years). Models are in large extent

calibrated against very few years of limnological records so far (Soares and Calijuri, 2021), potentially limiting the robustness of hindcast/ forecast exercises on long-term timescales such as pluri-decadal or pluri-centennial.

      To address this limitation, the first aim of our study is to adapt an existing modeling approach for long-term studies, constrained by climate scenarios of the IPCC, and to calibrate and validate the model against up to 63 years of limnological records from four perialpine lakes monitored by the Observatory of LAkes (OLA) (Rimet et al., 2020): Geneva, Annecy,

Bourget, and Aiguebelette. Here we evaluate the possibility to force mechanistic models by following the long-term evolution of shortwave radiation (W.m$^{-2}$) and air temperature (°C) while providing realistic seasonal trends for the other parameters, such as wind speed, for which local scale downscaling often lacks accuracy. We use unique long observation datasets to insure the robustness of our findings.

      The second objective of this study is to investigate the evolution of the thermal regime and the solubility of oxygen,

as a biological indicator, in the lakes over 1850-2100 to assess their different responses and sensitivity to global warming.

      We begin 1) by describing our approach for long-term forecast/ hindcast based on a reduced number of meteorological forcing variables, then 2) by presenting the model calibration and validation over a relatively short period of 10 years, using both complete and reduced meteorological inputs, then 3) we test the method over longer timescales, i.e., against 37 to 63 years of monitoring data, and finally 4) we explore trends in lake thermal regime through 15 physical indices, and we estimate

effects of climate change on the dissolved oxygen in the four lakes over the 1850-2100 period using an ensemble of climate projections based on the Shared Socioeconomic Pathways (ssp126, ssp370, and ssp585) (Riahi et al., 2017a).

**Table 1**. Characteristics of the four study sites

|  | Geneva | Annecy | Bourget | Aiguebelette |
|---|---|---|---|---|
| Location | 46°27'; 6°32' | 45°86'; 6°17' | 45°43'; 5°52' | 45°33'; 5°48' |
| Mean/Max depth (m) | 154/309 | 41/65 | 85/145 | 31/70 |
| Area (km$^2$) | 581.3 | 27.59 | 44.5 | 5.45 |
| Mean elevation (m.a.s.l.) | 372 | 447 | 232 | 390 |
| Residence time of water (yrs) | 11.3 | 3.8 | 9 | 3.1 |
| Mixing regime | mono-mero-mictic | monomictic | monomictic | monomictic |
| Trophic state | Mesotrophic | Oligotrophic | Oligo-mesotrophic | Oligo-mesotrophic |



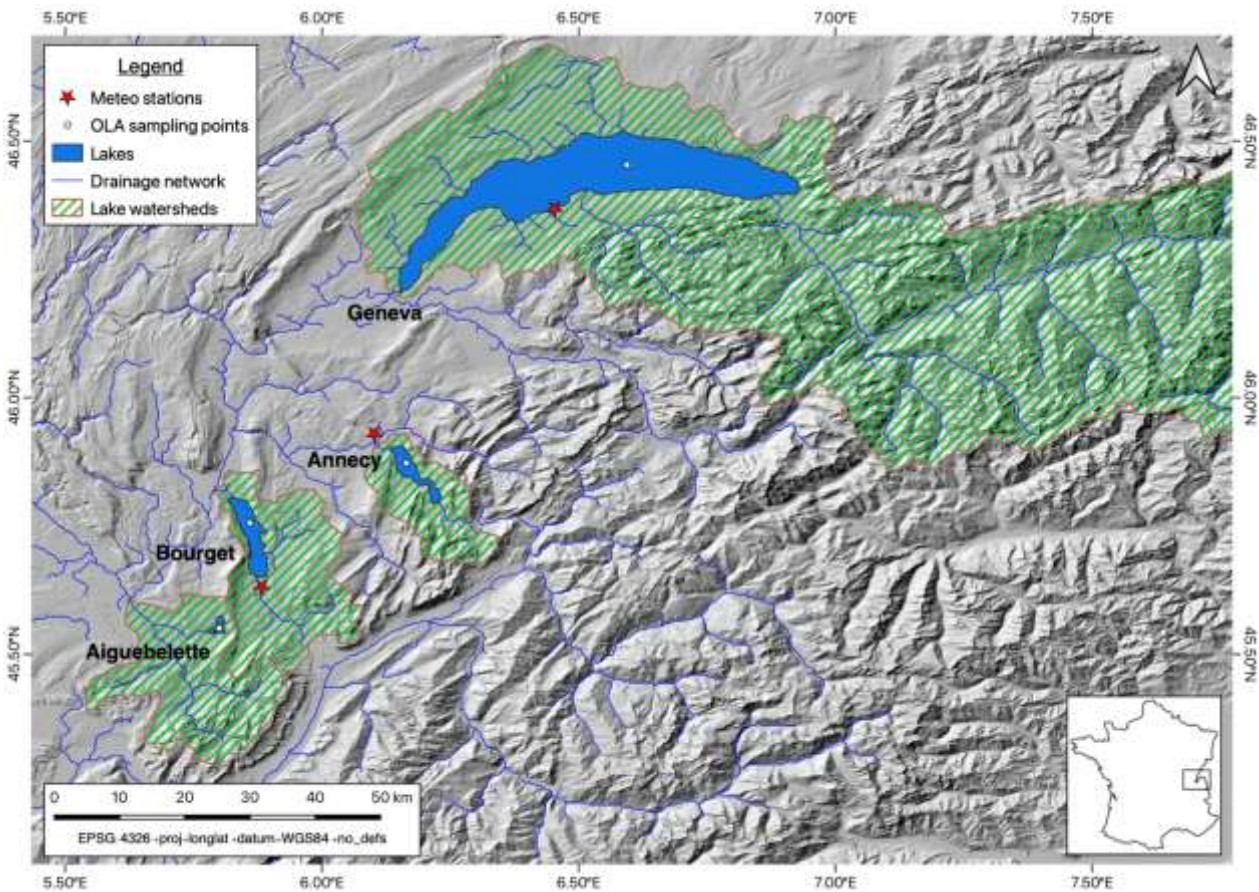

**Figure 1. Location of the study sites in peri-alpine region**. OLA sampling sites (Rimet et al., 2020), locations of meteorological stations, drainage watersheds, and networks are represented.

## 2 Methods

### 2.1 Study sites

We consider four lakes located in the peri-alpine region in France: Lake Geneva, Lake Annecy, Lake Bourget, and Lake Aiguebelette (Figure 1). They are situated in a continental mountain climate, and less than 150 km separate the four lakes from each other. The four lakes are monomictic – they mix only once per year –, deep and of glacial origin. Lake Geneva and Lake Aiguebelette are mesotrophic lakes, whereas Lake Annecy and Lake Bourget are oligotrophic and oligo-mesotrophic, respectively (Table 1).



**2.2 Lake data**

The Observatory of LAkes (OLA) managed by the CARRTEL (https://si-ola.inra.fr/si_lacs/login.jsf) has regularly monitored the physical and chemical conditions of peri-alpine lakes. Limnological data derived from the OLA observatory databases provides several decades (from 37 to 63 years) (Table 5) of monitoring on lake thermal conditions, i.e., water temperatures along the water column in the pelagic zone. The study sites were monitored for up to six decades by CARRTEL. Measurements were recorded at the deepest point of each lake, generally twice a month from March to November and once a month during the rest of the year, except for Lake Aiguebelette which is monitored 6 times a year.

**2.3 Hydrodynamic modelling**

A modeling approach was performed to run five one-dimensional hydrodynamic lake models simultaneously (FLake, GLM, GOTM, Simstrat and MyLake) to simulate the vertical water temperature profile in the four perialpine lakes. The same configuration and driver data were used for each lake to account for different sources of uncertainties in the model predictions. We used the R package LakeEnsemblR version 1.0.0. (Moore et al., 2021). The models were calibrated and validated against OLA limnological data, and further used to assess long-term trends in the thermal regime of each lake. The MyLake model was selected to develop and test our approach for long-term reconstruction that extends the simulation period beyond the instrumental one, as this model is well adapted to Northern and alpine regions (Couture et al., 2018; Kobler and Schmid, 2019; Saloranta, 2006). MyLake was developed by the Norwegian Institute for Water Research (NIVA), the University of Helsinki (Finland) and Université Laval (Canada). It simulates daily vertical water temperature profiles, density stratification, seasonal ice and snow cover, sediment-water dynamics, and phosphorus-phytoplankton interactions (Saloranta and Andersen, 2007).

**2.4 Climate warming scenarios**

The model was forced with statistically bias-adjusted (Lange, 2019a) and downscaled climate projections (Cucchi et al. 2020; Lange 2019b) (ISIMIP3BASD method) from phase 3b of the Inter-Sectoral Impact Model Intercomparison Project (ISIMIP3b). This was based on the output of phase 6 of the Coupled Model Intercomparison Project (CMIP6 (Eyring et al., 2016)). The better-performing models, providing daily data for all variables during the period of interest (1850-2100) were selected (Lange, 2019a) (GFDL-ESM4, IPSL-CM6-LR, UKESM1-0-LL, MPI-ESM1-2-HR and MRI-ESM2-0 (Table S2)) and compared to meteorological station data.

All five climate models were downscaled at 0.5° (55km) resolution. These models are good representatives of the entire CMIP6 ensemble as they are characterized with low (GFDL-ESM4, MPI-ESM1-2-HR, MRI-ESM2-0) and high climate sensitivity (IPSL-CM6A-LR, UKESM1-0-LL). Air temperature and downwelling shortwave radiation from each model were compared to available observation data from a meteorological station located at Thonon-les-bains (1987-2019 and 1971-2019 for air temperature and shortwave radiation, respectively) (Monitoring data from the INRAE CLIMATIK platform



(https://agroclim.inrae.fr/climatik/, in French) managed by the AgroClim laboratory of Avignon, France). Years with data collected over a period less than 350 data collected were not considered. Annual means and daily means during the monitoring periods were calculated and compared between modelled and observed data. Three error assessment metrics (root mean squared error – RMSE, normalized mean absolute error – NMAE and Pearson correlation coefficient – r) were calculated to evaluate the performance of climatic models. The most performant of the 5 models (IPSL-CM6A-LR) was selected and used as input data for the study.

CMIP6 experiments used were historical climate from 1850 to 2014, and scenarios ssp126 (SSP1-RCP2.6 climate), ssp370 (SSP3-RCP7 climate), and ssp585 (SSP5-RCP8.5 climate) from 2015 to 2100. The SSPs consider how societal choices will affect greenhouse emissions, SSP1 being the most sustainable scenario and SSP5 the worst one (Riahi et al., 2017b). The Representative Concentration Pathways (RCPs) corresponds to the range of the year 2100's radiative forcings values, from 2.6 to 8.5 W m$^{-2}$ (van Vuuren et al., 2011).

**2.5 Meteorological forcing data**

The meteorological forcing variables required in the *LakeEnsemblR package* are air temperature (°C), downwelling shortwave radiation (W m$^{-2}$), 10-m elevation wind speed (m s$^{-1}$), cloud cover fraction, relative humidity (%), rain (mm.day$^{-1}$), and surface-level barometric pressure (Pa), at a daily time step.

From the chosen climate model (IPSL-CM6-LR), all forcing variables were extracted for the grid cells containing the four lakes, Lakes Bourget and Aiguebelette being situated in the same grid. A sensitivity test was carried out on all seven climate variables to validate our hypothesis that forcing variables can be reduced into the hydrodynamic models to only air temperature and shortwave radiations to identify long-term trends accurately (Table S1).

Then, simulations with the MyLake model for the four lakes were computed with four different climatic configurations: (ii) with only air temperature and shortwave radiation from ISIMIP3b while all other variables were extracted from meteorological observations from which daily means were calculated and replicated every year from 1850 to 2100; (iv) with all input meteorological forcing variables extracted from ISIMIP3b. The period of meteorological observations extends from 2000 to 2011 for Lake Geneva (MétéoSuisse Data Warehouse) and from 1959 to 2020 for the other lakes (Table 4) (SICLIMA database, INRAE climate series provider using Meteo France standard SAFRAN grid). Decoupling meteorological parameters is a strong assumption, which is addressed in the discussion section. The cloud cover was available only for Lake Geneva, and values were adopted as the same for the other lakes. Surface pressure was considered constant in this study. Configurations (i) and (iii) are based on configurations (ii) and (iv) respectively, with a correction factor for both air temperature and shortwave radiation (Table 2). These two variables were compared to available data from the closest meteorological stations (from INRAE and Meteo France networks – CLIMATIK and SAFRAN database), encompassing 32 to 61 years of meteorological time series data (Fig. 1) (Table 3). Correction factors were calculated from the difference between raw climatic model data and observed data from local stations, at the daily resolution, to fit better to meteorological data and correct the altitude bias (Fig. S2; Fig. S3). Further, daily corrected meteorological data were used between 1850 and 2100.



**Table 2. Meteorological forcing configuration and performance metrics** for MyLake (ISIMIP) calculated between daily simulations and observations. Outputs comparison between 4 configurations: (i) only air temperature (T°C) and shortwave radiations (Sw) with a correction factor, (ii) only air T°C and Sw without a correction factor, (iii) all ISIMIP3b forcing data with a correction factor applied to air T°C and Sw, and (iv) all ISIMIP3b forcing data, over a ten-year period.

| Configuration | Inputs variables | Lake | RMSE | Pearson_r | Bias | MAE | NSE |
|---|---|---|---|---|---|---|---|
| (i) | Air T°C and Sw with correction factor | Geneva | **1.143** | **0.961** | **0.302** | **0.727** | **0.918** |
| | | Annecy | 1.951 | 0.938 | -0.992 | 1.270 | 0.824 |
| | | Bourget | **1.022** | **0.963** | **-0.046** | **0.620** | **0.926** |
| | | Aiguebelette | **1.587** | 0.949 | **0.305** | **1.138** | **0.896** |
| (ii) | Air T°C and Sw No correction factor | Geneva | 1.583 | 0.957 | -1.02 | 1.557 | 0.842 |
| | | Annecy | 2.686 | 0.933 | -1.930 | 1.989 | 0.667 |
| | | Bourget | 1.389 | 0.960 | 0.902 | 1.110 | 0.863 |
| | | Aiguebelette | 2.290 | **0.950** | 1.825 | 2.012 | 0.764 |
| (iii) | ALL forcing data with correction factor | Geneva | 2.144 | 0.949 | 1.551 | 1.619 | 0.710 |
| | | Annecy | **1.699** | 0.942 | **0.331** | 1.277 | **0.867** |
| | | Bourget | 1.532 | 0.958 | 1.055 | 1.300 | 0.834 |
| | | Aiguebelette | 2.050 | **0.948** | 1.296 | 1.655 | 0.827 |
| (iv) | ALL forcing data No correction factor | Geneva | 1.398 | 0.955 | 0.739 | 1.028 | 0.877 |
| | | Annecy | 1.972 | **0.945** | -0.882 | **1.215** | 0.821 |
| | | Bourget | 2.233 | 0.948 | 1.887 | 2.015 | 0.646 |
| | | Aiguebelette | 3.200 | 0.945 | 2.760 | 2.878 | 0.576 |

**Table 3.** Meteorological data sources used to calculate the correction factors applied to Air temperature (T°C) and Shortwave radiations, for the 4 study sites.

| | Lake Geneva | Lake Annecy | Lake Bourget | Lake Aiguebelette |
|---|---|---|---|---|
| Data base | CLIMATIK - INRAE | CLIMATIK – Meteo France | SAFRAN | SAFRAN |
| Station | Thonon-les-Bains | Meythet | Voglans | Voglans |
| Period | Air T°C : 1987-2019 | Air T°C : 1993-2019 | Air T°C : 1959-2020 | Air T°C : 1959-2020 |
| | Shortwave : 1971-2019 | Shortwave : 2010-2017 | Shortwave : 1959-2020 | Shortwave : 1959-2020 |

Daily meteorological data from the SAFRAN analysis system were extracted from SICLIMA data base (Pagé, 2008), collected by Meteo France since January 1st of 1959. Data have been re-calculated from daily local observations at smaller grid cells

(8km x 8km) by Meteo France.

**Table 4.** Meteorological data sources used for the meteorological patterns calculated from daily means of Cloud cover (Cl), Relative humidity (Rh), Wind speed (Ws) and Rain (R) variables, for the 4 study sites.

| | Lake Geneva | Lake Annecy | Lake Bourget | Lake Aiguebelette |
|---|---|---|---|---|





| Data base | MétéoSuisse | SAFRAN (Rh,Ws,R) | SAFRAN (Rh,Ws,R) | SAFRAN (Rh,Ws,R) |
|---|---|---|---|---|
| | | MétéoSuisse (Cl) | MétéoSuisse (Cl) | MétéoSuisse (Cl) |
| Station | Nyon/Changins | Meythet | Voglans | Voglans |
| | | Nyon/Changins | Nyon/Changins | Nyon/Changins |
| Period | 2000-2011 | 1959-2020 (Rh,Ws,R) | 1959-2020 (Rh,Ws,R) | 1959-2020 (Rh,Ws,R) |
| | | 2000-2011 (Cl) | 2000-2011 (Cl) | 2000-2011 (Cl) |

## 2.6 Model set-up, calibration, and validation

MyLake was run through the package *LakeEnsemblR package*. The initial water temperature profile was extracted from a winter temperature profile of the OLA database. Air temperature from 1st to 31st January 1850 was compared to air temperature of the instrumental period to identify years with similar climate conditions. The water temperature profile of the year with the lowest RMSE between winter air temperatures (1992 for Lakes Bourget and Geneva - 2000 for Lakes Annecy and Aiguebelette) was used as the initial profile, 31st of January 1850.

The model was calibrated considering the most sensitive parameters identified in previous studies (Saloranta, 2006): scaling factors for wind speed and shortwave radiation and physical C_shelter parameter. The Latin Hypercube Calibration (LHC) method was used for calibration. The LHC method uses upper and lower bounds for all parameters considered, and then samples evenly within the parameter space given by these bounds. Then MyLake was run and evaluated for 100 parameters sets. The performance of the model was assessed through six statistical metrics: RMSE, Nash-Sutcliffe efficiency

(NSE), r, bias, MAE and NMAE.

        The optimal values of model parameters were determined based on the performance metrics. First, calibration and validation were performed over a period of 10 years, depending on the density of observation data for eack lake. This period corresponds to the temporal scale generally covered by modeling studies performed between 2015 to 2020 (Soares and Calijuri, 2021) (Table 5). Further, the robustness of the model to perform long-term simulations was assessed by computing the

195 performance metrics over the complete period with field data availability, covering 63, 54, 37, and 46 years for Lakes Geneva, Annecy, Bourget, Aiguebelette, respectively (Table 5).

**Table 5.** Monitoring, calibration and validation periods for the four peri-alpine lakes

| | Geneva | Annecy | Bourget | Aiguebelette |
|---|---|---|---|---|
| Monitoring | 1957-2020 | 1966-2020 | 1984-2021 | 1974-2020 |
| Calibration | 1980-1990 | 1970-1980 | 1998-2008 | 1995-2005 |
| Validation | 1990-2000 | 2000-2010 | 2008-2018 | 2005-2015 |
| Long-term validation | 1957-2018 | 1966-2020 | 1984-2018 | 1974-2020 |





### 2.7 Assessment of lakes response to warming

Model fit was assessed for 15 thermal indices: lake temperature (full profile), surface temperature (Ts=5m), bottom temperature (Tb=60, 60, 299 and 140 for Lakes Annecy, Aiguebelette, Geneva, and Bourget, respectively), mean temperature along the water column (Tm), metalimnion depth (top and bottom) defined as the water stratum with the steepest thermal gradient, demarcated by the bottom of the epilimnion and top of the hypolimnion (Wetzel, 2001b), epilimnion temperature, hypolimnion temperature, Schmidt stability (Idso, 1973), buoyancy frequency (Brunt-Vaisala frequency), thermocline depth and stratification characteristics (start, end, duration, maximum intensity and day of the year) (See supplementary). The R package rLakeAnalyzer (Winslow et al., 2019) was used to calculate epilimnion, hypolimnion, metalimnion, Schmidt stability, buoyancy frequency and thermocline depth. The dates of stratification onsets/break-ups were defined as the day when the surface-to-bottom temperature differences were greater or less than 2 °C (Robertson and Ragotzkie, 1990) for at least 5 consecutive days. The maximum stratification intensity was defined as the greatest difference between Ts and Tb. Water volumes above certain thresholds of ecological interest (7 °C, 9 °C and 12 °C) (Wolfe, 1996) above which the reproduction, growth, and survival of certain fish species may be affected (Mari et al., 2016b; Réalis-Doyelle, 2016) were calculated from bathymetry files. Finally, potential oxygen solubility was calculated as a function of water temperature following Winkler Table.

### 2.8 Statistical analysis

The slope of the significant trends was evaluated by least-squares linear regression and t-tests when residuals followed a normal distribution. Otherwise, the non-parametric Mann-Kendall test and the Theil-Sen method were used to estimate if the slope was significant and provide the slope's value. The Shapiro-Wilk test and Fisher's F test evaluated the distribution normality and variance homoscedasticity, respectively. Average values of each thermal metric in present (1990-2020) and future (2070-2100) were compared with either the Student t mean difference test in case of residuals normal distribution and homoscedasticity or the Welsh t mean difference test when variances were different. When normal distribution was not followed, the Mann-Whitney U test (equal variances) or the Kolmogorov-Smirnov test (different variances) were used instead. A p-value of 0.05 was used to represent the significanceof the statistical tests. Kernel densities were calculated to compare averaged daily data distribution over 2000-2010 and 2090-2100. The coefficient of overlap of the two distributions was calculated (Ridout and Linkie, 2009). The analyses were carried out with the R software (version 4.1.2, R Core Team, 2021).

## 3 Results

### 3.1 Model performance

The MyLake model, forced by air temperature and shortwave radiation, reproduced well the observed temperature along the water column in the four deep alpine lakes (Fig. 2). Model performances were compared across a 10-year validation and more extended periods (Table 6), depending on the availability of observations for each lake. During both validation




periods (i.e., 10 years and the 37-63 years period), the model predicted water temperature with good precision, as RMSE values are generally less than 2 °C and 1.22 °C for the two deepest lakes (Geneva and Bourget). The model reproduced well the inter-annual temperature variability of the four lakes (Fig. 3) with Pearson correlation coefficient values (r) >0.9 over the 10-year calibration and validation periods. The model robustness has been maintained over the long term as r values remained >0.9.

Depending on the lake considered, the model either slightly underestimated the water temperature (bias = -0.03-0.05 °C and -0.91-0.99 °C for Lake Bourget and Annecy, respectively) or overestimated it (+0.31-+0.69 °C and +0.3 °C for Lakes Aiguebelette and Geneva respectively). These results showed that this is not a systematic tendency to over or underestimate the water temperature however depends on the lake characteristics.

The ability of the model to predict the evolution of specific thermal indices has been assessed (Table 7). The model

performance in predicting epilimnion temperatures was the best for the two deepest lakes (Geneva and Bourget), with RMSEs ranging between 1.92 and 2.08 °C. For Lake Annecy, the discrepancies obtained for the epilmnion simulations were similar to those of the simulated temperature profiles over the 10-year validation period (RMSE=2.02 and 1.56 °C, respectively). Still, it was less effective over the long-term validation (RMSE=3.69 °C). Similarly, the model was more performant in predicting the epilimnion temperature of Lake Aiguebelette during the 10-year validation period (RMSE=2.61 °C) compared to the long-

term validation (RMSE=4.6 °C). The epilimnion inter-annual variability of the four lakes was well reproduced, with $R^2$>0.89 and $R^2$>0.65 for the 10-year and long-term validations respectively.

A clear difference in amplitude has been identified between Schmidt stability calculated from simulations and observed water temperature profiles (mean RMSE = 3270.7, 4092.6, 3391.9, 1967.1 for Geneva, Bourget, Annecy, and Aiguebelette, respectively) however general seasonal patterns across the four lakes were well simulated by the model ($R^2$>0.86

for the 10-year and $R^2$>0.66 for the long-term validation periods). The average model performance between the two validation periods after comparing calculations of observed and simulated thermocline depth is found to be better for the 2 deepest lakes (with RMSE calculated as a percentage of lakes total depths =6.7 %, 8 %, 13.6 %, and 18 % for lakes Bourget, Geneva, Aiguebelette and Annecy, respectively). The RMSE associated with the prediction of the onset stratification was lower for Lake Bourget and Aiguebelette over the long-term validation period (RMSE=12.9 and 15.8 days, respectively) and less

accurate for Lake Annecy and Geneva (RMSE=28.0 and 20.9 days, respectively). The estimation of the end-of-stratification date was reasonably close for Lake Bourget and Annecy (10.7<RMSE<13.4 days and 15.2<RMSE<18.9 days for 10-year and long-term validation periods respectively). The model predicted better the end-of-stratification for Lake Aiguebelette (RMSE=5.5/13.2 days) compared to Lake Geneva (RMSE=17.0/23.7 days). Moreover, the stratification duration was represented more accurately for Lake Bourget (RMSE=13.09 and 17.7 days for the 10-year and long-term validation periods,

respectively) and ranges between one month and one month and a half for the other lakes (25.9<RMSE<41.9 days).

**Table 6**. **MyLake performance indicators of water temperature simulations** (root mean squared error – RMSE and Pearson correlation coefficient – r) for the calibration and validation periods (val=10 years; lt-val=long-term); n $_{obs\ data}$ = number of limnological observation data for comparison with simulated data.



| Lake | Period | RMSE (°C) | r | $n_{obs\,data}$ |
|---|---|---|---|---|
| Geneva | cal | 1.143 | 0.961 | 3868 |
| | val | 1.107 | 0.967 | 3756 |
| | lt-val | 1.211 | 0.961 | 17126 |
| Annecy | cal | 1.723 | 0.944 | 1048 |
| | val | 1.952 | 0.938 | 1185 |
| | lt-val | 1.949 | 0.938 | 5231 |
| Bourget | cal | 1.022 | 0.963 | 31847 |
| | val | 1.105 | 0.970 | 6508 |
| | lt-val | 1.119 | 0.958 | 93127 |
| Aiguebelette | cal | 1.587 | 0.949 | 10901 |
| | val | 1.589 | 0.933 | 19830 |
| | lt-val | 1.608 | 0.941 | 40699 |


**Table 7. Comparison of model validation metrics for eight thermal indices** over 10 years and a long-term validation period (37 to 63 years) for the four lakes (see extended table in supplementary) Bold = RMSE<2°C and $R^2$>0.7; Italic = RMSE<2.7°C and $R^2$>0.5.

| | | RMSE | | $R^2$ | |
|---|---|---|---|---|---|
| | Lake | 10 years | *Long-term | 10 years | *Long-term |
| Full profile - water temp | Geneva | **0.79** | **0.67** | **0.87** | **0.89** |
| | Annecy | **1.56** | **1.84** | **0.86** | **0.75** |
| | Bourget | **0.79** | **1.02** | **0.93** | **0.88** |
| | Aiguebelette | **1.48** | **1.85** | **0.86** | **0.74** |
| Epilimnion T°C | Geneva | **1.92** | *2.08* | **0.91** | **0.87** |
| | Annecy | *2.02* | 3.69 | **0.9** | *0.67* |
| | Bourget | *2.01* | **1.9** | **0.90** | **0.90** |
| | Aiguebelette | *2.61* | 4.60 | **0.89** | *0.65* |
| Hypolimnion T°C | Geneva | **0,68** | **0.57** | 0.15 | 0.18 |
| | Annecy | **0.84** | **0.9** | 0.1 | 0.13 |
| | Bourget | **0.51** | **0.80** | 0.15 | 0.01 |
| | Aiguebelette | **1.31** | **1.19** | 0.14 | 0.09 |
| Schmidt Stability (J/m^2) | Geneva | 3108,40 | 3433.10 | **0.89** | **0.87** |
| | Annecy | 820.9 | 5962.9 | **0.86** | *0.69* |
| | Bourget | 3993.60 | 4191.6 | **0.89** | **0.88** |
| | Aiguebelette | 1536.03 | 2398.21 | **0.88** | *0.66* |
| Thermocline depth (m) | Geneva | 28.00 | 21.20 | 0.27 | 0.22 |





| | | | | |
|---|---|---|---|---|
| Annecy | 11.7 | 11.8 | 0.22 | 0.11 |
| Bourget | 7.92 | 11.5 | *0.69* | 0.37 |
| Aiguebelette | 8.80 | 9.43 | *0.56* | 0.25 |

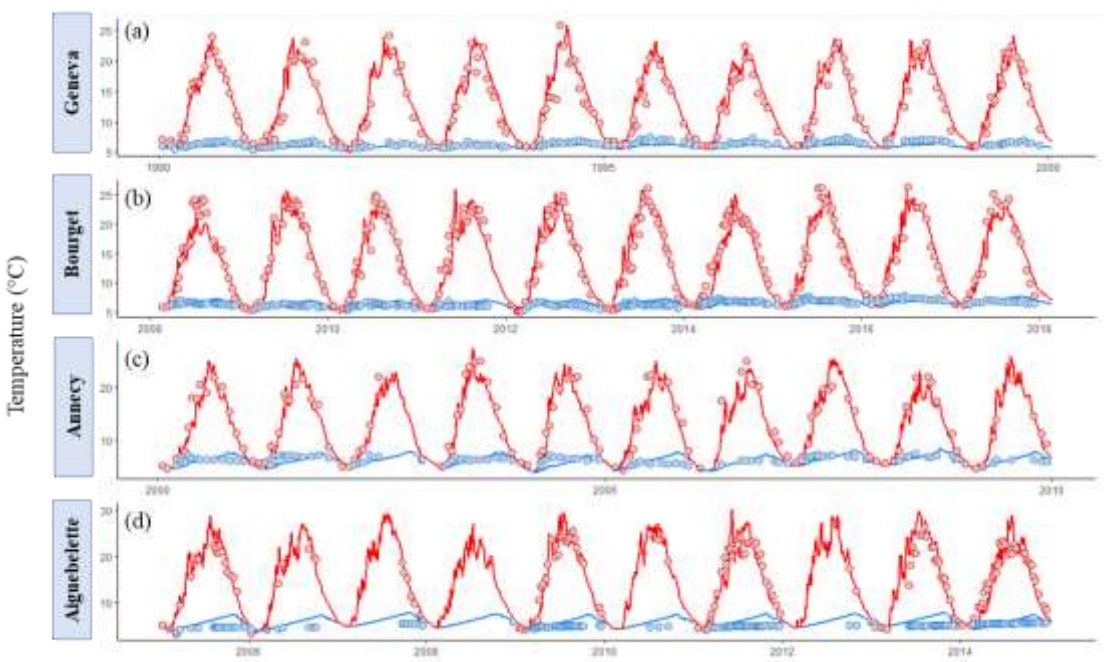


**Figure 2. Temporal variations in epilimnion (red) and hypolimnion (blue) temperatures** of Lake Geneva (a), Bourget (b), Annecy (c) and Aiguebelette (d). MyLake simulations (line) vs Observations (point) over the 10-year validation period.





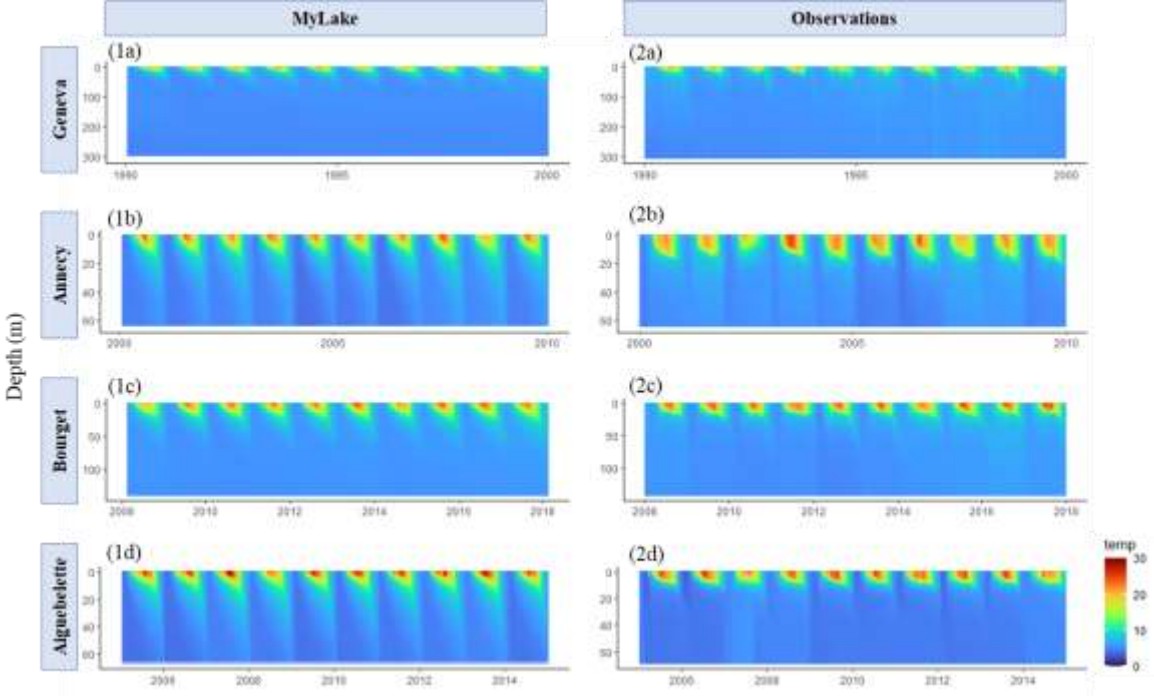

**Figure 3. Daily averaged MyLake water temperature simulations** (1) **and interpolated observations from OLA database**
(2) in Lake Geneva (a), Annecy (b), Bourget (c) and Aiguebelette (d) over the 10-year validation period.

## 3.2 Meteorological trends

Based on the different scenarios adopted in the present work, mean annual air temperature has increased by +0.39 °C
(Lake Bourget) and +0.5 °C (Lake Geneva) per decade over the past 30 years (from 1990 to 2020) (Fig. 4). At the horizon
2100 (from 2070 to 2100), on one hand the mean annual air temperature with the ssp126 scenario decreased on average by -
0.008 °C per decade across all lakes. On the other hand, with the ssp370 and ssp585 scenarios an average increase of +0.9 °C
and +1.13 °C per decade were predicted, respectively. Mean annual shortwave radiations have increased on average by +1.58
W m$^{-2}$ per decade from 1990 to 2020 and are expected to decrease by -1.5 W m$^{-2}$ per decade according to the most optimistic
scenario (ssp126). This result is consistent with a previous study, showing that the increasing trends in solar radiation are not
likely to continue in future (Schmid and Köster, 2016). An increase of +0.6 W m$^{-2}$ to +1.87 W m$^{-2}$ could be expected according
to the ssp370 and ssp585 scenarios, respectively.





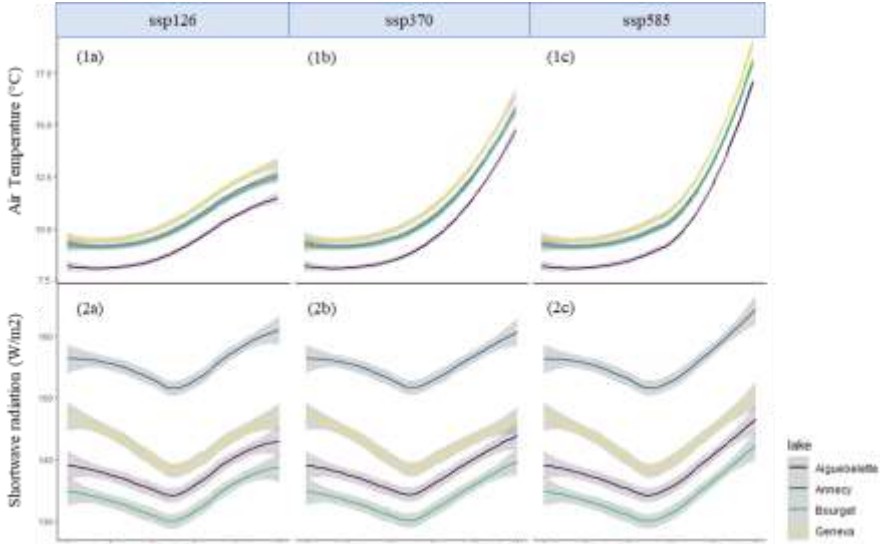

**Figure 4. Selected meteorological forcing variables over studied period 1850-2100.** Projected evolution of air temperature (1) and shortwave radiation (2) from IPSL-CM6A-LR (ISIMIP3b) under ssp126 (a), ssp370 (b) and ssp585 (c)for the four lakes (Geneva, Annecy, Bourget, Aiguebelette), from 1850 to 2100.

### 3.3 Lakes response to the meteorological scenarios

### 3.3.1 Water temperature

The epilimnion temperature increased by around 0.44 °C decade$^{-1}$ (Lake Aiguebelette) to 0.48 °C decade$^{-1}$ (Lake Geneva) over the past 30 years (1990-2020). In the future projections (2070-2100), according to the most optimistic scenario (ssp126), a decrease of -0.07 to -0.08 °C decade$^{-1}$ could be expected (Fig. 5). The intermediate scenario (ssp370) predicted a significant increase of +0.77 °C decade$^{-1}$ (Lake Annecy) to +0.89 °C decade$^{-1}$ (Lake Geneva). In the worst-case scenario, the epilimnion temperature in the four lakes could increase by +1.03 °C decade$^{-1}$ (Lake Annecy) to +1.13 °C decade$^{-1}$ (Lake Geneva). In all cases, a significant change in epilimnion temperature was predicted by the model over the two periods.

Over the last 30 years, hypolimnion temperature increased by +0.29 °C, +0.31 °C, +0.32 °C and +0.39 °C decade$^{-1}$ in Lake Aiguebelette, Geneva, Annecy, and Bourget, respectively. The rate of increase was higher for surface layers than for deep hypolimnetic layers. The difference between the epilimnion and hypolimnion warming rates was larger for Lake Geneva. A less significant increase in hypolimnion temperature could be expected at the horizon 2100 according to the ssp126 scenario, with +0.19 °C decade$^{-1}$ (Lake Geneva) to +0.3 °C decade$^{-1}$ (Lake Bourget). Unlike the epilimnion, even the most optimistic scenario predicted an increase in deep layers. In the four lakes, a significant rise in +0.55 °C decade$^{-1}$ (Lake Annecy) to +0.73 °C decade$^{-1}$ (Lake Bourget) was expected in the case of the intermediate scenario. The ssp585 scenario predicted an increase by +0.67 °C decade$^{-1}$ (Lake Annecy) to +0.82 °C decade$^{-1}$ (Lake Geneva).



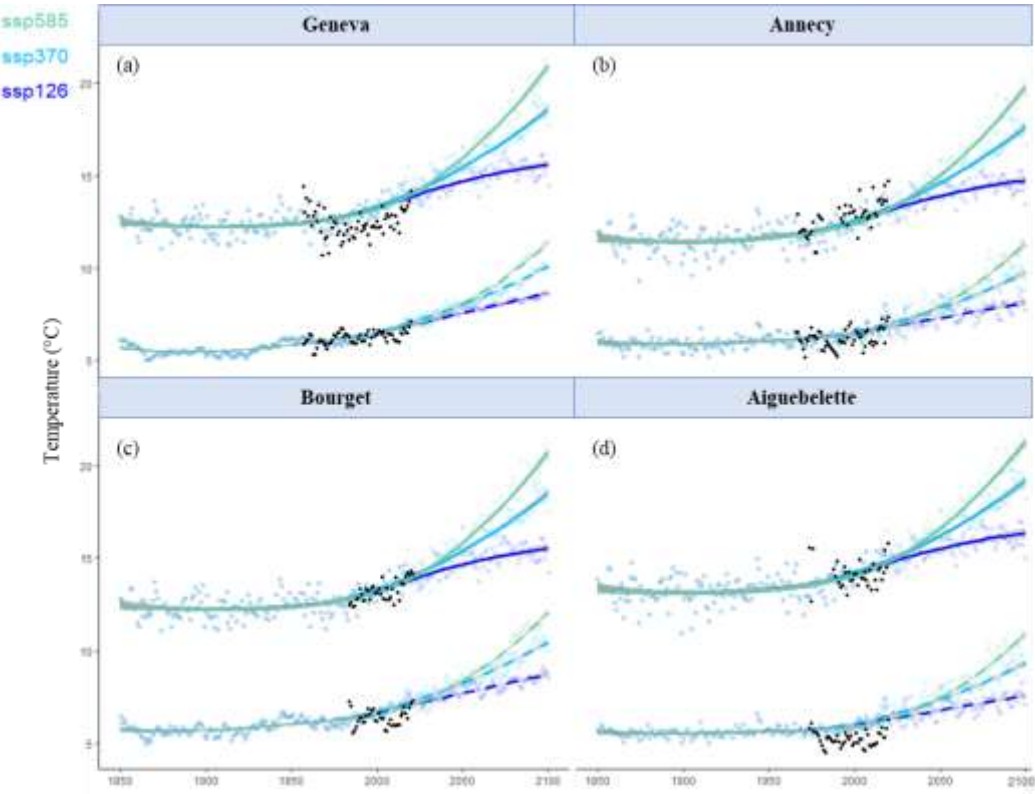

**Figure 5. Annual averages of epilimnion (line) and hypolimnion (dashed lines) temperatures** from MyLake daily water temperature simulations over the period 1850-2100, for three different climate scenarios (ssp126, ssp370, ssp585) in Lake Geneva (a), Annecy (b), Bourget (c), and Aiguebelette (d). Black dots represent annual averages of observation data.

The water temperature change was quantified as the non-overlapped area of the two daily averaged temperature distributions in the present (2000-2010) and the future (2090-2100) as a percentage of the combined area of those distributions for the intermediate scenario (ssp370) (Fig. 6). The greatest thermal change occurred in Lake Geneva and Bourget with 90 % and 86 % non-overlap respectively between the two periods. In Lake Annecy and Aiguebelette, 77 % and 76 % thermal non-overlap were predicted, respectively.

The annual average temperature was expected to increase between the two periods on average by +3.59, +3.60, +3.66 and +3.92 °C in Lake Annecy, Geneva, Aiguebelette and Bourget respectively.





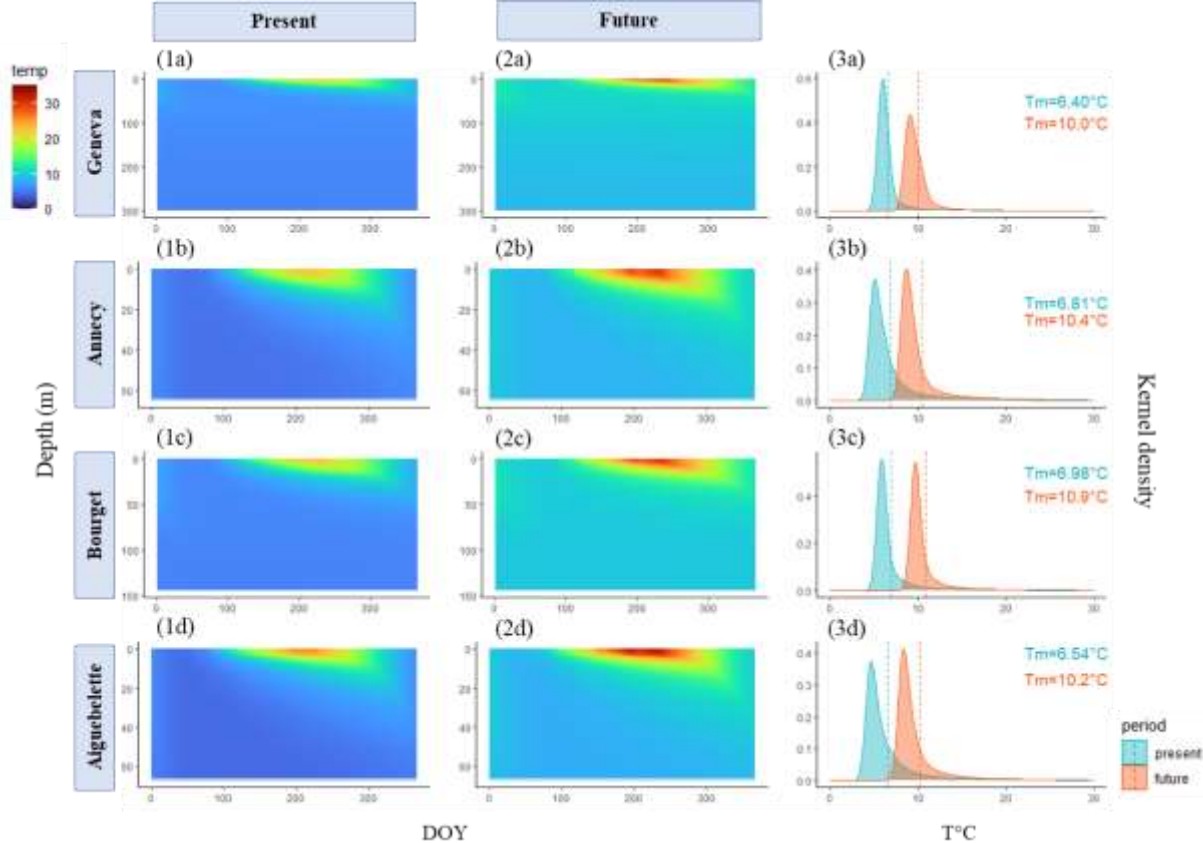

**Figure 6. MyLake water temperature simulations** from intermediate climate scenario (ssp370). Daily averaged water temperature over the periods 2000-2010 (1) and 2090-2100 (2) in Lake Geneva (a), Annecy (b), Bourget (c), and Aiguebelette (d). Estimated frequency of daily average temperature data (3) over these present (2000-2010) and the future (2090-2100) using kernel density. The dashed lines represent average annual temperatures over the entire periods.

### 3.3.2 Stratification characteristics

Schmidt stability, describing the stability of the water column and its resistance to mixing, has significantly increased over the past 30 years by an annual average of +174.6 J m$^{-2}$ decade$^{-1}$, +522.97 J m$^{-2}$ decade$^{-1}$, +654,57 J m$^{-2}$ decade$^{-1}$, and +1753.5 J m$^{-2}$ decade$^{-1}$ for Lake Annecy, Aiguebelette, Bourget and Geneva respectively (Fig. 7). No significant trend was predicted by the model at the horizon 2100 for the ssp126 scenario in Lake Geneva and Bourget. For Lake Annecy and Aiguebelette, a significant decrease of -100.5 J m$^{-2}$ decade$^{-1}$ and -281.3 J m$^{-2}$ decade$^{-1}$e could be expected according to that optimistic scenario, respectively. The evolutions predicted by the model in the case of the intermediate scenario varied between the four lakes, with an increase of +285.5 J m$^{-2}$ decade$^{-1}$ in Lake Annecy, +927.8 J m$^{-2}$ decade$^{-1}$ in Aiguebelette, +1088.7 J m$^{-2}$ decade$^{-1}$ in Bourget, and +3435 J m$^{-2}$ decade$^{-1}$ in Lake Geneva. In the worst-case scenario (ssp585), an increase of +445 J m$^{-2}$ decade$^{-1}$ (Lake Annecy) to +4695 J m$^{-2}$ decade$^{-1}$ (Lake Geneva) could be expected. For the four lakes, in any scenario,





a significant evolution was predicted by the model between the last 30 years and the future (2070-2100), however with different annual averages from one lake to another (Schmidt stability=2646 J m$^{-2}$ and 23042 J m$^{-2}$ for Lake Annecy and Geneva respectively).

No significant trend was identified for the day of onset of stratification (DOY) in the present and the future, except for Lake Annecy according to ssp126 scenario (DOY =+1.6 days decade$^{-1}$ and +4.3 days decade$^{-1}$ for present and future periods respectively) (Fig. 7). For Lake Geneva, the onset of stratification occurred significantly earlier in the case of ssp585 by on average 11 days on average (DOY=96 and 85 in the present and the future respectively), contrary to the ssp126 scenario, which predicted a significant delay of 5 days later (DOY=96 to 101 for present and future periods respectively). No significant change in stratification onset has been identified for Lake Annecy, Bourget and Aiguebelette.

In regards to the break-up of stratification, a significant trend was only predicted for Lake Geneva in the present (1990-2020) with an average delay of 5 days decade$^{-1}$ later. Except for Lake Annecy and Aiguebelette ssp126 and Lake Bourget ssp370, the end of stratification appeared significantly later in the future than in the last 30 years. In the ssp126 scenario, the stratification could end on average 6 days and 5 days earlier for Lake Geneva and Bourget, respectively. According to the ssp370 scenario, it was predicted to end on average 5.7 days, 7.4 days and 3 days later in Lake Annecy, 350 Geneva and Aiguebelette, respectively. In the worst-case scenario, a delay of 7.7 days, 14.7 days, 6.8 days and 3.6 days in Lake Annecy, Geneva, Bourget and Aiguebelette, respectively.

As a result, a significant increase in the average stratification duration was predicted in Lake Geneva – ssp585 of +21.2 days (DOY$_{present}$=278.3 and DOY$_{future}$=299.5). In the most optimistic scenario (ssp126), a significant decrease could be expected in Lake Geneva and Bourget (-8.8 days and -9.45 days, respectively) between the two periods. No significant 355 difference was predicted by the model for Lake Annecy and Aiguebelette. The average duration of stratification within the four lakes was from 246 days (Lake Annecy) to 279 days (Lake Geneva) for the present period. In the worst-case scenario, the duration could last from 251 days (Lake Annecy) to 300 days (Lake Geneva).



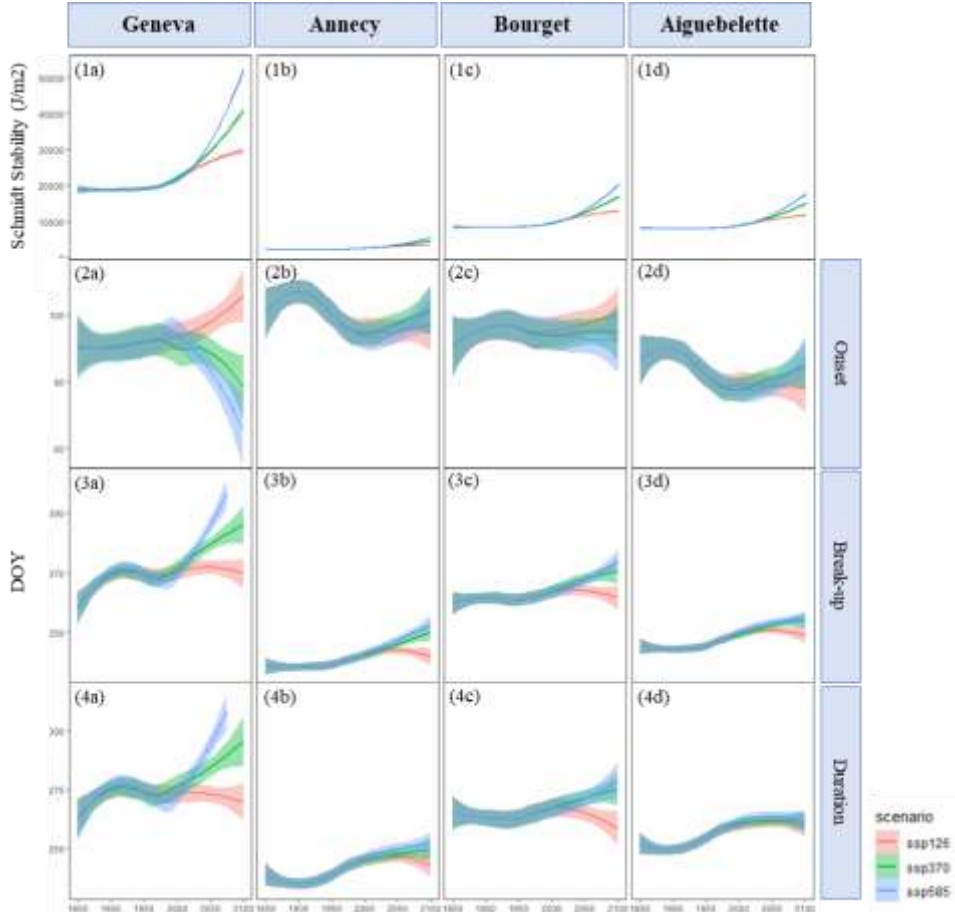

**Figure 7. Stratification trend characteristics for the four lakes** (Geneva (a), Annecy (b), Bourget (c) and Aiguebelette (d)).
Estimations of Schmidt stability (1) from June to September, stratification onset (2), break-up (3), and duration (4) over the
period 1850-2100 for the 3 scenarios (ssp126, ssp370, and ssp585), calculated from MyLake water temperature simulations.

### 3.3.3 Water volumes: habitat

Changes in thermal habitat between the present (2000-2010) and the future (2090-2100) were assessed based on the
lake volume fraction that exceeded specific temperature thresholds (>7 °C, >9 °C, and >12 °C) (Fig. 8), above which the
reproduction, growth, and survival of certain fish species may be affected (Mari et al., 2016b; Réalis-Doyelle, 2016). It was
then quantified as the non-overlapped area of the two lake volume fraction distributions (present and future based on the 3
different scenarios). In Lakes Geneva and Bourget, temperature was predicted to exceed 7 °C in all case scenarios, with a lake
volume fraction non-overlap of 100 % between the present (2000-2010) and the future (2090-2100). In Lake Annecy, 77.8 %
of the water column could reach a temperature greater than 7 °C in the ssp126 case-scenario and could extend to the entire



water column in the two other scenarios, with a 100 % increase between present and future projections. In Lake Aiguebelette, the lake volume fraction with temperature above 7 °C has increased from 15 % (present) to 87 % (ssp370 and ssp585), involving a lake volume fraction non-overlap of 49 % (ssp126) and 100 % (ssp370 and ssp585). In all lakes, a significant increase of water volume above 7 °C was predicted by the model for the three scenarios.

In Lake Annecy, Geneva, and Bourget, temperature could exceed 9 °C according to the intermediate scenario (annual average lake volume fraction=70 %, 84 %, and 100 % respectively). According to the ssp585 scenario, temperatures of almost the entire lakes could reach 9 °C (annual average lake volume fraction=96-100 %). The specific bathymetry of Lake Aiguebelette caused a progressive increase of lake volume fraction with temperature above 9 °C, yet never reached the total volume (72.5 % of the lake for ssp585). According to the ssp126 scenario, lake volume fraction with temperature above 9°C

non-overlap was 35 %, 47 %, 61 %, and 62 % for Lake Aiguebelette, Annecy, Bourget and Geneva respectively. The increase was more important for the two larger and deeper lakes. In all cases, the model predicted a significant change between present and the three future scenarios. The thermal overlaps between the present and the future according to ssp585 scenarios were predicted to reach 100 % for all four lakes. This means that the average daily lake volume fraction with temperature above 9 °C was expected to be completely different between the two periods, with higher values in the future. A 100 % increase was

also predicted in the ssp370 case-scenario for Lake Geneva and Bourget, against 69 % and 75 % for Lakes Aiguebelette and Annecy, respectively.

       Water volumes above 12 °C have increased by on average by +45 % (ssp126) for Lakes Annecy, Geneva, and Bourget, with a slightly smaller increase in Lake Aiguebelette (+30 % of lake volume fraction non-overlapped). The model predicted the same increase in the ssp370 case scenario for the two deepest lakes (Geneva and Bourget) with on average +70

% of lake volume fraction non-overlap. In Lake Aiguebelette and Annecy, +53 % and +34 % could be expected, respectively. Finally, in the worst-case scenario (ssp585), a higher increase in water volume above 12 °C was predicted for Lake Geneva (+91 % of non-overlap), followed by Lake Bourget (+82 % of non-overlap), Lake Annecy (+70 % of non-overlap) and Lake Aiguebelette (+64 % of thermal non-overlap).



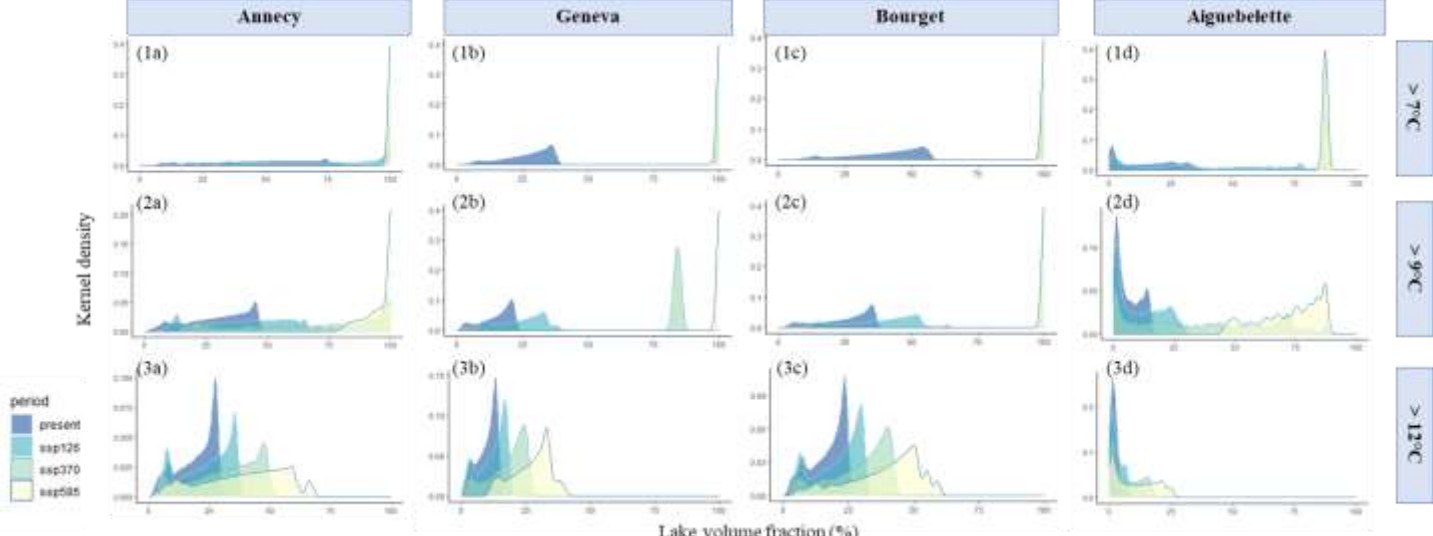

**Figure 8. Lake volume fraction with temperature exceeding three characteristics thresholds** (7 °C (1), 9 °C (2), and 12 °C (3)) calculated from simulated water temperature profile and bathymetry in Lake Annecy (a), Geneva (b), Bourget (c), and Aiguebelette (d). Kernel density estimates of daily average lake volume fraction over the present (2000-2010) and the future (2090-2100) predicted by the three scenarios (ssp126, ssp370 and ssp585).

### 3.3.4 Oxygen solubility

Oxygen solubility was calculated from the Winkler tables, as a function of temperature. In all scenarios, significant trends in potential oxygen solubility were predicted by the model for the four lakes. Over the last 30 years (1990-2020), an average annual decrease of -0.07, -0.09, -0.12, and -0.13 mg $L^{-1}$ decade$^{-1}$ was computed for Lake Geneva, Aiguebelette, Annecy, and Bourget, respectively (Fig. 9). At the horizon 2100 (2070-2100), according to the ssp126 scenario, a decrease of -0.05 mg $L^{-1}$ decade$^{-1}$ (Lakes Geneva and Aiguebelette) to -0.09 mg $L^{-1}$ decade$^{-1}$ (Lake Bourget) could be expected. A significant decrease of -0.16 mg $L^{-1}$ decade$^{-1}$ (Lake Geneva) to -0.21 mg $L^{-1}$ decade$^{-1}$ (Lakes Bourget and Aiguebelette) was predicted in the intermediate scenario and from -0.17 mg $L^{-1}$ decade$^{-1}$ (Lake Annecy) to -0.23 mg $L^{-1}$ decade$^{-1}$ (Lake Bourget) in the ssp585 scenario. In all cases, a significant difference has been forecast between present and future periods within the four lakes. An annual average of 12.15, 12,21, 12.22, and 12.3 mg $L^{-1}$ was calculated for the present period from the model simulations in Lakes Bourget, Annecy, Geneva, and Aiguebelette, respectively. In the future predicted by the intermediate scenario, annual averages from 11.28 mg $L^{-1}$ (Lake Bourget) to 11.49 mg $L^{-1}$ (Lake Aiguebelette) could be expected.

Furthermore, the model predicted a faster decrease in oxygen solubility in the epilimnion compared to the hypolimnion in the four lakes over the last 30 years, with an average of -0.104 and -0.096 mg $L^{-1}$ decade$^{-1}$ in the epilimnion and the hypolimnion, respectively. No significant trend was forecast for oxygen solubility in the epilimnion for all lakes




according to the ssp126 scenario, resulting in the stabilization of oxygen solubility. In the ssp370 case-scenario, the model predicted a decrease in oxygen solubility in the epilimnion of -0.16, -0.17 and -0.18 mg L$^{-1}$ decade$^{-1}$ in Lake Annecy, Aiguebelette and both Geneva and Bourget, respectively. According to the worst scenario (ssp585), oxygen solubility in epilimnion was expected to decrease by an average of -0.2 mg L$^{-1}$ decade$^{-1}$ (±0.01 mg L$^{-1}$ decade$^{-1}$). Oxygen solubility in deep layers are expected to be closely related to its evolution into the epilimnion, depending on the intensity and depth of the water

column mixing.

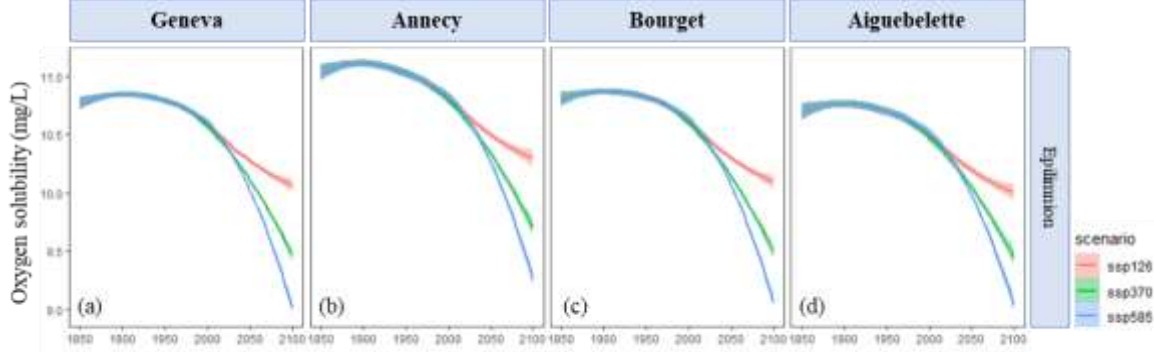

**Figure 9. Annual averages of potential oxygen solubility over the period 1850-2100**, in the epilimnion calculated from MyLake daily water temperature simulations for three different climate scenarios (ssp126, ssp370, ssp585) in Lake Geneva (a), Annecy (b), Bourget (c), and Aiguebelette (d).


Potential change in oxygen solubility was also quantified as the non-overlapped area of the two daily averaged oxygen solubility distributions in the present (2000-2010) and the future (2090-2100) as a percentage of the combined area of those distributions, for the intermediate scenario (ssp370) (Fig. 10). As for the thermal regime, the highest changes in oxygen solubility were in Lakes Geneva and Bourget with 60 % of non-overlap between the present and the future. In Lakes

Aiguebelette and Annecy, the non-overlap rate has been reduced to 55 % and 54 %.

Annual potential average oxygen solubility was predicted to decrease by -0.9 mg L$^{-1}$ (Lake Geneva) to -1.1 mg L$^{-1}$ (Lake Bourget) in the future.

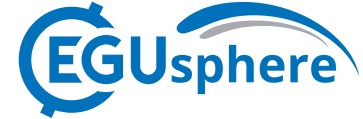

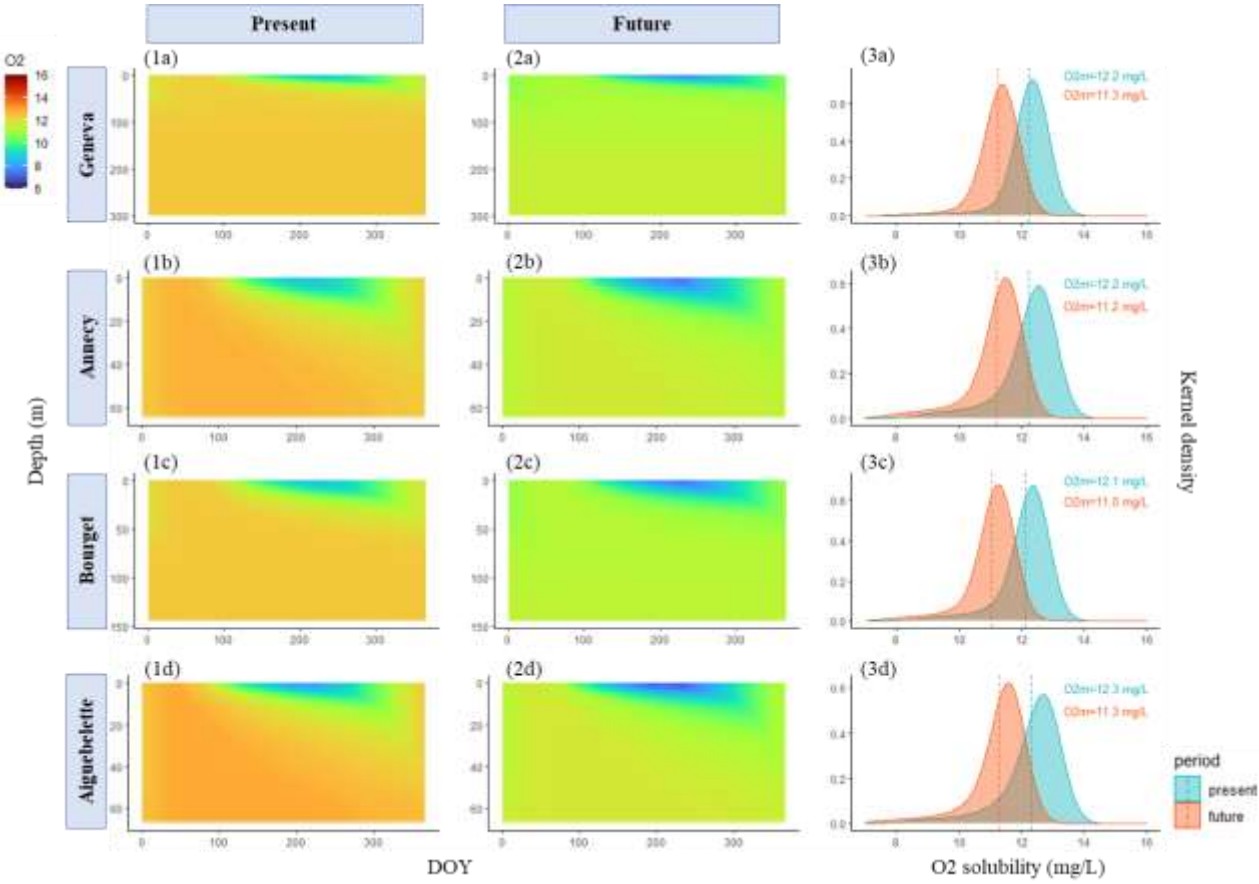

**Figure 10. Potential oxygen solubility in lake waters of the four lakes**, calculated from MyLake water temperature simulations for the intermediate climate scenario (ssp370). Daily averaged oxygen solubility over the present (1) and future (2) in Lakes Geneva (a), Annecy (b), Bourget (c), and Aiguebelette (d). Estimated frequency of daily average oxygen solubility (3) over the present (2000-2010) and future (2090-2100) using kernel density. The dashed lines represent annual average oxygen solubility over the entire periods.

As for the thermal habitat, water volumes with sufficient dissolved oxygen levels to support fish survival were assessed based on lake volume fraction that exceeded certain thresholds (10 mg L$^{-1}$ and 11 mg L$^{-1}$) (Fig. 11). The differences between the present (2000-2010) and the future (2090-2100) according to the three scenarios were then quantified as the non-overlapped area of the two lake volume fraction distributions. In the present, an average of 95.6 % (±1.6 %) of the total lake volume had a potential oxygen solubility>10 mg L$^{-1}$ in Lakes Annecy, Geneva, and Bourget, unlike Lake Aiguebelette with only 61.2 % of the lake volume. Averages of 25.8 % (±3.2 %), 40.5 % (±3.5 %) and 48.3 % (±5.3 %) of lake volume fraction non-overlap was predicted in ssp126, ssp370 and ssp585 scenarios, respectively. In the worst-case scenario, lake volume




fractions were expected to be reduced to 85.9 %, 92 %, 86.8 % and 57.7 % for Lakes Annecy, Geneva, Bourget and Aiguebelette respectively.

In the two deepest lakes (Geneva and Bourget), the lake volume fraction with potential oxygen solubility above 11 mg L$^{-1}$ has decreased gradually according to the 3 scenarios, and reached 100 % of non-overlap between the present and the future (ssp585). The decrease was less sharply for Lakes Annecy and Aiguebelette, with 72 % and 68 % of non-overlap in ssp585 scenario, respectively. Although, in the intermediate scenario, 68.7 % (Geneva), 67.7 % (Annecy), 61.6 % (Bourget), and 48.6 % (Aiguebelette) of the total lake volume could have potential dissolved oxygen levels greater than 11 mg L$^{-1}$. These

lake volume fractions could be reduced to 52.3 %, 41.3 %, 33.2 %, and 26.8 % in Lakes Annecy, Aiguebelette, Geneva, and Bourget respectively, in the worst-case scenario.

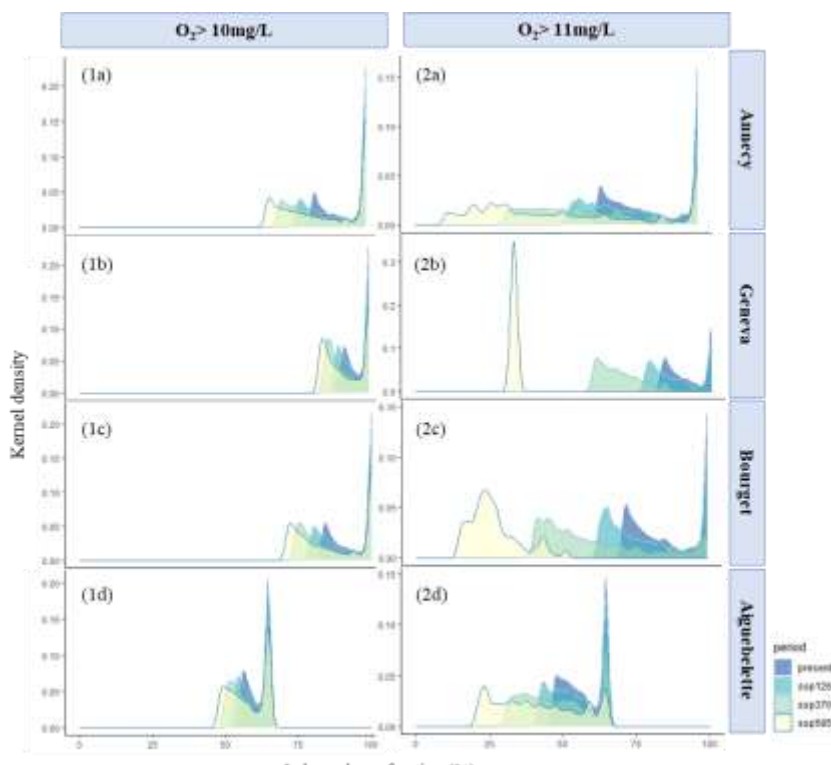

**Figure 11.** Lake volume fraction with oxygen solubility exceeding two thresholds (10 (1) and 11 mg L$^{-1}$ (2)), calculated from
simulated water temperature profiles and bathymetry in Lake Annecy (a), Geneva (b), Bourget (c) and Aiguebelette (d). Kernel density estimates of daily average lake volume fraction over the present (2000-2010) and the future (2090-2100) predicted by the three scenarios (ssp126, ssp370 and ssp585).



## 4. Discussion

### 4.1 Warming trends

Over the last 30 years and according to the results presented in this study, the epilimnion temperature increased on average by +0.46 °C decade$^{-1}$ in the four lakes. This rate agrees well with warming rates of surface water temperature found for other alpine lakes, over the period 1975-5015 (Ficker et al., 2017). It is consistent with the average increase in air temperature (from 0.39 to 0.5 °C per decade), indicating a direct response in lake temperature trends. Moreover, the water temperature has increased faster in the epilimnion than in the deep layers of the hypolimnion, with a considerable difference in the warming rate of increase between lakes. The hypolimnion temperature has increased between +0.29 (Lake Aiguebelette) and +0.39 °C decade$^{-1}$ (Lake Bourget). The difference in increase between the upper and deep layers was smaller for Lake Bourget ($\Delta$Raise decade$^{-1}_{epi-hypo=}$ +0.07 °C decade$^{-1}$) compared to Lake Geneva, Annecy and Aiguebelette ($\Delta$Raise decade$^{-1}_{epi-hypo=}$+0.15-0.16 °C decade$^{-1}$). Therefore, the increase in epilimnion temperature over the last 30 years was quite similar between the 4 perialpine lakes. However, Lake Bourget has experienced a higher temperature increase in its deep layers compared to the three other lakes. Although large and deep lakes with greater heat storage capacities are expected to show rising hypolimnetic temperatures due to their high potential of heat carryover from one year to the next, the lower deep warming in Lake Geneva could be explained by its greater wind exposure that enhances deeper winter mixing of the water column leading to the plunging of cold water from melting glaciers in deep layers.

Temperatures from the last 30 years were compared to those from the future period 2070-2100. Differences between the two periods show that the water column temperature responses to global warming is highly variable across scenarios. In the most optimistic scenario, where strong environmental and political measures would be implemented from today, the response of upper and lower layers was predicted to be opposed in the four lakes, with a decrease in temperature in the surface layers (-0.07-0.08 °C decade$^{-1}$) and an increase in the deep layers (+0.19-+0.3 °C decade$^{-1}$). This difference in response could be explained by the thermal inertia and heat accumulation in deep waters, which has already been observed in response to seasonal variations (Crossman et al., 2016). Furthermore, the decrease of surface layers temperature in future has already been shown, under certain climate projections (Schmid and Köster, 2016).

In the intermediate (ssp370) and most pessimistic (ssp585) scenarios, epilimnion and hypolimnion temperatures are increasing. The increase in epilimnion temperature is slightly lower in the case of Lake Annecy (+1.03 °C decade$^{-1}$) and Lake Aiguebelette (+1.04 °C decade$^{-1}$). In both scenarios, the most important increase in epilimnion temperature was predicted for Lake Geneva (+0.89 °C for ssp370 and +1.13 °C decade$^{-1}$ for ssp585), yet it remained very close to the increase in epilimnion temperature in Lake Bourget (+0.86 °C decade$^{-1}$ and +1.11 °C decade$^{-1}$, respectively).

As for the epilimnion temperature, the smallest hypolimnion temperature increase was predicted for Lake Annecy according to the ssp370 and ssp585 scenarios (+0.55 °C and +0.67 °C decade$^{-1}$ respectively), with very close values from Lake Aiguebelette. The most significant increase could be expected in Lake Bourget for the intermediate scenario (+0.73 °C decade$^{-1}$) and in Lakes Geneva and Bourget for the most alarming scenario (+0.82 °C decade$^{-1}$ and +0.81 °C decade$^{-1}$, respectively).



The evolution of water temperature was quite similar across the four lakes in the case of the most optimistic scenario where global warming would tend to decrease. However, the model predicted differences in response for the four lakes in the two other scenarios. In future projections, the water column of the two shallowest lakes (Annecy and Aiguebelette) could warm up less quickly than the other two deeper lakes (Geneva and Bourget), which is different from recent observations. This might be related to a high wind exposure leading to a deeper mixing of the water column in those deeper lakes. Although, in both recent and future periods in case of the two pessimistic scenarios (ssp370 and ssp585), epilimnion temperature was expected to increase faster than hypolimnion temperature, unlike in the ssp126 scenario.

Finally, we expect a higher thermal change in Lakes Bourget and Geneva, comparing the entire water column temperature between present and future periods according to the intermediate scenario, with an average increase of +3.92 °C and +3.6 °C, respectively. However, by 2100, there is a high probability that the hydrological regime in the Rhône River upstream Lake Geneva changes because of an earlier snow melting leading to an earlier, and maybe shorter, input of cold water into the lake. This change could have an effect on the thermal regime and oxygen solubility in Lake Geneva.

## 4.2 Oxygen trends

Over the last 30 years, oxygen solubility as a function of water temperature has decreased in the four lakes, with differences in amplitude. Indeed, the most significant decrease was observed for Lakes Annecy and Bourget (-0.12/-0.13 mg L$^{-1}$ decade$^{-1}$), in contrast to Lake Aiguebelette with a -0.07 mg L$^{-1}$ decade$^{-1}$ decrease. These results provides information on the potential oxygen solubility relative to the evolution of the water column temperature, however they do not integrate oxygen consumption and production by photosynthesis. Our approach allows us to assess the likely evolution of oxygen solubility in a global warming context. Still, it would need to be adjusted with studies dealing ecosystem oxygen production and consumption in the pelagic zone. For instance, the decrease in oxygen solubility can reduce habitat for cold water fish, who could face warmer water temperature and lower dissolved oxygen concentrations (Mohseni et al., 2003). Rates of decrease in oxygen solubility are similar to those observed in temperate lakes (Jane et al., 2021). Moreover, a faster decrease in the epilimnion has been identified, which was correlated to a higher increase in the water temperature of surface layers.

For the future projections, according to the most optimistic scenario (ssp126), a similar decrease in potential oxygen solubility was expected for Lakes Geneva and Aiguebelette (-0.05 mg L$^{-1}$ decade$^{-1}$) and Lakes Annecy and Bourget (-0.08/-0.09 mg L$^{-1}$ decade$^{-1}$). Lake Bourget was predicted to experience the greatest decline according to all three scenarios, which was consistent with the water temperature increase along the water column. In the worst-case scenario, Lake Geneva and Bourget would face the greatest decrease in oxygen solubility, which is consistent with previous observations on thermal changes. However, the predicted decreases in oxygen solubility were very close within the four lakes, with no significant differences (from -0.16 to -0.21 mg L$^{-1}$ decade$^{-1}$ in the ssp370 scenario and from -0.17 to -0.23 mg L$^{-1}$ decade$^{-1}$ in the ssp585 scenario).





When comparing change in daily average oxygen solubility during one year between the present (2000-2010) and the future (2090-2100), the decrease was similar between the Lakes Geneva and Bourget (with 60 % and 61 % of non-overlap) and slightly less important in Lakes Annecy and Aiguebelette (49 % and 55 % of non-overlap). Once again, the two deepest lakes were expected to face the most important decrease in oxygen solubility. Furthermore, the annual average oxygen solubility was very close in the four lakes in both periods ($O_{2annual\ mean}$ = 12.2 mg $L^{-1}$ and $O_{2annual\ mean}$ = 11.2 mg $L^{-1}$ in the present and future respectively).

Finally, the effect of decreased oxygen solubility on fish habitats was assessed through some potential thresholds. The model predicted faster decreases of the lake volume fraction with oxygen solubility>11 mg $L^{-1}$ in the two deepest lakes (Geneva and Bourget), according to the three scenarios.

## 4.3 Sensitivity of peri-alpine lakes to climate change

Despite the similarity in typology (large, deep), the four studied perialpine lakes showed different responses to global warming. For instance, the highest increase of Schmidt stability was predicted in Lake Geneva according to ssp370 and ssp585 (+3435 and +4695 J $m^{-2}$ decade$^{-1}$, respectively), Lake Annecy being the least sensitive to the evolutions predicted by the three scenarios (+285 and +445 J $m^{-2}$ decade$^{-1}$ according to ssp370 and ssp585 respectively). These results seemed to be consistent as Lake Geneva was the most extensive, making it more vulnerable to higher shortwave radiations and air temperatures. Despite their different morphologies and depths, Lakes Bourget and Aiguebelette were predicted to have similar Schmidt stabilities and identical responses to global warming in the three considered scenarios  (Schmidt stability = 9986 J $m^{-2}$ and 9224 J $m^{-2}$ over the period 1990-2020 for Lakes Bourget and Aiguebelette respectively). This observation may reveal a higher resistance to mixing in Lake Aiguebelette, considering its lower depth.

As for Schmidt stability, Lake Geneva was predicted to experience the highest changes in the start, end and duration of stratification, according to the 3 scenarios. Indeed, in future projections, according to the most optimistic scenario, the stratification occurred on average 11 days earlier according to the most alarming scenario. The later stratification onset seemed consistent as the air temperature was expected to stabilize or even decrease (-0.03 °C decade$^{-1}$ for Lake Geneva) in the ssp126 scenario. Similarly, shortwave radiations are expected to decrease in the most optimistic scenario, with an average of -1.2 W $m^{-2}$ (Lake Geneva) to -1.8 W $m^{-2}$ (Lake Annecy). No significant trend was predicted in the 3 other lakes during the two studied periods. However, Lake Annecy stratified the latest over the last 30 years (DOY=97) whereas Lake Aiguebelette stratified on average earlier (DOY=89). The model predicted stratification onsets to occur almost simultaneously (DOY=96 and 97 for Geneva and Bourget respectively).  Concerning the stratification end-date and duration, the model forecasts were very close for the two deepest lakes over the period 1990-2020 (DOY$_{end-of-stratif}$ = 11 and 1; Stratification duration = 279 and 270 days for Lakes Geneva and Bourget, respectively). On the contrary, stratification at Lake Annecy ended the earliest with a shorter duration of stratification during that same period (DOY$_{end-of-stratif}$ = 343 and Stratification duration = 245 days). In the future, Lake Geneva will be more vulnerable to global warming with later end-of-stratification date and longer stratification duration



(Stratification break-up = +7.4 and +14.7 days decade$^{-1}$ according to ssp370 and ssp585 respectively; Stratification duration =
+21.2 days decade$^{-1}$ in ssp585 case scenario). These results are coherent with the differential warming of surface and deep
waters, which could influence the strength and duration of a lake's stratification period (Råman Vinnå et al., 2021). Moreover,
the stratification lasted longer in the two deepest lakes over the two periods (e.g. Stratification duration = 300, 278, 251 and
237 days in Lakes Geneva, Bourget, Annecy and Aiguebelette respectively according to the ssp585 scenario). These changes

in the beginning, end and duration of stratification could also have an impact on oxygenation conditions in deep waters (Roberts
et al., 2009b), in addition to the oxygen solubility decreasing when water temperature increases.

        In regards to the change in thermal habitat and according to model simulations, the entire water column of Lake
Bourget and Geneva would exceed 7 °C in all 3 scenarios. Likewise, almost the entire water column could reach a temperature
above 9 °C in Lakes Geneva and Bourget, according to the ssp370 and ssp585 scenarios. A significant change was predicted

between the present and the future in all scenarios in the four lakes, with a gradual increase in the lake volume fraction with
temperatures exceeding  9 and 12 °C, with the two deepest and largest lakes (Geneva and Bourget) being more sensitive (e.g.
+91 % and +82 % increase between the present and the future -ssp585). In Lake Aiguebelette, a slower increase in lake volume
fraction with temperatures exceeding these thresholds could be expected according to the 3 different scenarios. This could be
explained by a shallower mixing of the water column due to a lower exposure to the wind.

Change in thermal habitat is could be an important issue for the lake managers and fishers, as it impacts the behavior
and reproduction of certain fish species. Previous studies have demonstrated that Arctic charr (*Salvelinus alpinus*), an
emblematic fish of perialpines Lakes (Caudron et al., 2014) has a limited thermal tolerance range compared with other
salmonids (Baroudy and Elliot, 1994; Elliot and Elliot, 2010) and hatch survival decreased significantly as the water
temperature at spawning increased from 5 °C to 8.5 °C (Mari et al., 2016b). Furthermore, trout survival has been shown to

decrease with an increase of +4 °C in water temperature (Réalis-Doyelle, 2016) and whitefish (Coregonus sp.) population, the
main targeted fish in periaplines lakes (Anneville et al., 2017)  will suffer of increased temperature (Gerdeaux, 2004 ; Eckmann,
2013).

**4.4 Model reliability and limitations of the approach for the long term**


        The approach developed in this study seems well adapted to long-term simulation approaches, with potential
implications for paleolimnological studies, long-term forecast studies, or studies focused on the effects of climate change on
lake ecological dynamics. The model errors for the long-term were relatively small for the 4 study sites, with RMSE < 2 °C
during ten-year calibration and validation periods. The model performed better in the long term with lower RMSE for the two

deepest lakes Geneva and Bourget (RMSE long term = 0.67 and 0.9 °C, respectively). For Lakes Annecy and Aiguebelette,
the RMSE were lower over the 10-year validation period but remained acceptable over the long-term with RMSE below 2 °C.
Thus, applying only air temperature and downwelling shortwave radiation from climatic projections provided a well-adapted
model to the study of the four perialpine lakes in the long term, even with considering only the seasonal trends in wind speed,



cloud cover, air relative humidity, and rainfall. In this sense, the approach seems adequate for long-term simulation approaches.
No systematic error has been identified as the model slightly underestimated (bias = -0.03-0.05 °C for Lake Bourget) or overestimated (bias=+0.3 °C for Lake Geneva) the water temperature, depending on the lake morphology. Positive biases could be attributed to the measurements which carried out around mid-day and in good weather, whereas model outputs were averaged daily (Vinçon-Leite et al., 2014). Metric errors were calculated for 15 variables to assess the ability of the model to reproduce them. As for the full profile, the model reproduced epilimnion temperature with better precision for Lakes Geneva
and Bourget. For Lakes Annecy and Aiguebelette, the model was a little less accurate on the long term but with good correlation coefficients, showing its ability to reproduce interannual variability. Hypolimnion temperature was simulated with better precision in Lakes Geneva and Bourget in comparison to the two shallower lakes (Annecy and Aiguebelette), with RMSE<0.7 °C, nevertheless the interannual variabilities were not well captured by the model ($R^2$<0.2). The model predicted with good precision the Schmidt stability seasonal variability for the four lakes, with greater accuracy for Lakes Bourget and Geneva
even on the long term ($R^2$>0.87). The thermocline depth was more difficult to predict by the model in Lakes Geneva, Annecy and Aiguebelette, yet good results were observed for Lake Bourget ($R^2$>0.67). These uncertainties could be caused in part by the presence of internal seiches, which increased the variability and made it difficult to be reproduced by 1-D models (Ayala et al., 2020).

The metric error calculation of the start, end and duration of stratification showed that the model could be used to
assess the long-term trends but the dates were to be considered with hindsight (R2<0.35 except for Lake Aiguebelette end-of-stratification). However, the best prediction was made for Lake Bourget, with averages RMSEs of 11.1, 13.2 and 15.8 days for onset, break-up and duration respectively.

This method allowed to overcome certain limitations such as the quality of the input files related to the climate variables scaling. For instance, the wind can be very different within a same 0.5° grid, depending no the surrounding terrain.
The influence of rivers and watersheds may also be difficult to simulate on the long-term. Reducing the number of forcing variables to only air temperature and shortwave radiations is an efficient approach to analyse thermal regime and oxygen solubility trends on the long-term. This could be applied to global forecast and paleolimnological studies.

A further limitation of this method is related to the correlation between the different climate variables, such as relative humidity and air temperature dependency. As the meteorological patterns replicated the seasonal fluctuations of each variable,
this error is limited. Another possible option would have been to use weather generator to simulate climate variables evolutions, implemented to integrate these correlations.

**5 Conclusion**

In this study, an approach to simulate long-term trends in lake thermal regime and oxygen solubility has been tested and validated against 63 years of limnological data from the OLA lakes observatory. The approache shows that 1-D thermal lake models perform well when run only with air temperatures and shortwave radiations as forcing variables, hence allowing

to overcome certain limitations related to the quality of climate input data for the long-term. Future application of the 1D model approach for long-term variations can be anticipated in the field of paleolimnology, but also to assess past and future effects of climate change on the ecological dynamics and lake habitats.

Simulations show that over the last 30 years, epilimnion temperature has increased on average by +0.46 °C decade$^{-1}$ in the four lakes. At the horizon 2100, simulations anticipate a minimum increase of +0.77 °C decade$^{-1}$ and +0.56 °C decade$^{-1}$ in epi- and hypolimnion respectively (Lake Annecy - ssp370) to a maximum of +1.13 °C decade$^{-1}$ (epilimnion) and +0.82 °C decade$^{-1}$ (hypolimnion) (Lake Geneva – ssp585). The response to climate change varies between lakes, with the deepest lakes (Bourget and Geneva) experiencing the fastest warming, with on average +3.92 °C and +3.6 °C at the horizon 2100.

In regards to oxygen condition, a decrease in oxygen solubility occurred over the last 30 years at least in Lakes Annecy and Bourget, with -0.12 mg L$^{-1}$ and -0.13 mg L$^{-1}$ decade$^{-1}$ respectively. At the horizon 2100, simulations indicate that Lakes Bourget and Geneva will face the greatest decrease of oxygen solubility, with -0.21 mg L$^{-1}$ decade$^{-1}$ according to the intermediate scenario, which may alter the chemical and ecological functioning of the lakes. Differences in the oxygen solubility response to climate change are also observed. Simulations of the duration and intensity of thermal stratification suggest that the decrease in lake oxygen conditions will be more pronounced in the case of Lake Geneva and that Lake Annecy would be the least sensitive to climate change.

**Code and data availability**

Code and data used in this paper are available from the corresponding author upon a reasonable request.

**Author contributions**

JPJ, OD, DB, PAD, BVL conceived and designed the study. OD performed the analyses, calculation and wrote the original draft of the paper. JPJ supervised the manuscript from the methodological and long-term trend analysis perspectives. All authors actively took part in the interpretation of the results and revisions of the paper.

**Competing interests**

The contact author has declared that neither they nor their co-author has any competing interests.

**Disclaimer**

**Acknowledgements**





We thank the HESS editor, the reviewers and the participants of the lively discussion in HESSD for their comments that helped
to improve the paper. We gratefully acknowledge the financial support from the ANR C-ARCHIVES and the Pole ECLA
(French Research Organization in Lake Ecology). We would like to thank the French Observatory of Lakes (OLA) for
providing the long-term limnological data and Victor Frossard from CARRTEL for thorough advances on the manuscript
conception.

670   **Financial support**

This research has been supported by the ANR C-ARCHIVES (grant no. 5283) and the Pole ECLA.

**Review statement**



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
