# Peer review of "Past and future climate change effects on thermal regime and oxygen solubility of four peri-alpine lakes"

_EGUsphere, 2022_

## Author Response (AR1)

Le Bourget-du-Lac, September 27th 2022

Dear Hydrology and Earth System Sciences Editors,

We would like to thank the *Hydrology and Earth System Sciences* Editors for giving us the opportunity to revise the manuscript, and the two reviewers for their constructive comments on our manuscript « *Past and future climate change effects on thermal regime and oxygen solubility of four peri-alpines lakes* » by O. Desgué-Itier, L. Soares, O. Anneville, D. Bouffard, V. Chanudet, P-A. Danis, I. Domaizon, J. Guillard, T. Mazure, N. Sharaf, F. Soulignac, V. Tran-Khac, B. Vinçon-Leite, J-P. Jenny. Please find enclosed a revised manuscript that takes into account the concerns raised by the Reviewers.

We carefully considered all comments, and accordingly, we added statements to clarify why only MyLake model was used instead of the other models. As recommended, the discussion section has been entirely revised in order to focus on the interpretation rather than the description of the results. A paragraph has also been included to the discussion to compare MyLake model with other ensemble model members. Besides, we have modified all figures to improve their quality, resolution and labels sizes. Finally, we improved the introduction and the method sections to clarify the rationale of our study and the reasons why we chose to use only shortwave radiation and air temperature projections for model hindcasts and future projections.

Please find below in blue text our responses to the comments made by the two Reviewers.

Sincerely yours

Olivia Desgué-Itier
"Ingénieur d'Etudes"
Now at CARRTEL Limnology Center, France
olivia.desgue@inrae.fr

[Figure]

Response to the reviewers' comments

Reviewer #1:

Comments:

Major points:

The abstract (L25: "[...] several 1D lake model's [...]") as well as the introduction and methods part make the impression that the study will be mainly about running an ensemble of lake models on peri-alpine lakes, but in the manuscript only MyLake results are shown and interpreted (all other ensemble model output are in the Supplementary). Here, I think several points are missing: (1) an explicit statement explaining why MyLake was used instead of the other ensemble members, (2) a revision of the text to make it clearer that the study is highlighting MyLake, and (3) a discussion paragraph looking at other ensemble model members. To (1), table S4 is showing that MyLake has the best model fits, and it makes sense to use it. I think this should be stated instead of L115 "[...] as this model is well adapted to Northern and alpine regions" as this is also true for Simstrat which was explicitly developed and applied on Alpine Lake systems. Further, Simstrat also incorporates TKE production by seiche which was mentioned as a shortcoming of the current study in replicating thermocline depth dynamics (L612). Further, the Supplementary is missing information on which model parameters were calibrated for FLake, GLM, GOTM and Simstrat. Also, as the title suggest the focus on oxygen dynamics, why wasn't for example GLM chosen to simulate these dynamics directly instead of MyLake which in the LakeEnsemblR version does not quantify dissolved oxygen dynamics at all?

We want to thank Reviewer #1 for his/her many suggestions to improve our paper. Following suggestions, we clarified wording: 1) a statement has been completed to clarify why MyLake was selected among the 5 models (L 119-122), 2) we precised that MyLake was the model selected for the trend analysis both in the abstract (L 30) and introduction (L 88), and 3) a paragraph comparing the five models was added to the discussion (L 655-664). A Table has been added to the Supplementary (Table S6) with the model parameters calibrated for each model. Concerning the ability of Simstrat model to incorporate TKE production by seiche and GLM model to simulate oxygen dynamics, that's a very good point. Over the five different models, MyLake had an overall better performance during our limited calibration exercise. Yet this study does not intend to rank the models. Here, we made the choice to select only the best performant model for the four lakes, based on the lower RMSE's.

Comments made in the PDF have been addressed and are highlighted in blue in the manuscript. The line numbering refers to the new version of the manuscript (c.f. Draft_HESS_version_RC1_RC2.docx).

Resolution of figures: the resolution of most figures (2, 3, 4, 5, 6, 7, 8, 9, 10, 11) is very low and their axes labels are not readable. Please improve the resolution.

All figures have been saved in .pdf for a higher quality. The font size of the axis text has been increased for better readability.

L479: This sentence seems wrong to me. First, wouldn't increasing density gradients especially in deep lakes reduce vertical mixing and prevent hypolimnetic water temperatures from increasing. Further, Lake Geneva's surface area is big (which I suppose you mean here by wind exposure) but the wind energy itself as well as the reach of TKE is supposed to be decreasing in the future (see Castro et al 2021, Nature Comm Earth&Env). Second, how should enhanced deep winter mixing (assuming you refer to convective overturn) lead to plunging of cold water from melting glaciers in deep layers? Is this referring to the Rhone River receiving increased meltoffs? But in that regard, increased mixing would lead to increased deep water temperatures and that would cause a colder Rhone inflow to entrain in to deeper water layers.

Institut National de Recherche pour l'Agriculture, l'Alimentation et l'Environnement (INRAE) - http://www.inrae.fr
CARRTEL Limnology Center, 75 bis, avenue de Corzent – CS 50511,  F-74203 Thonon-les-Bains cedex,  France

[Figure]

We appreciate the arguments raised here. Accordingly, we added the explanation suggested and the reference to the literature Castro et al. 2021 (L 505-509). We also deleted the sentence in relation to the cold water coming from melt glaciers plunging into deep layers as inflows temperature was considered as constant in that study.

L502: The statement that the simulated warming is different from current observations due to high wind exposure is not enough in my opinion. Couldn't this be a model shortcoming and bias. Are you sure only the wind mixing pathway is the main difference here? Please focus more on discussing why the future projections are different from recent observations.

Thank you for pointing this out. We implemented the discussion on the potential causes of the different responses between future projections and recent observations on Lines 510-526. No temporal bias was exhibited in the error metrics over time. The difference might be explained by less and less frequent complete mixing in deeper lakes, leading to longer inter-mixing periods during which water temperature increases. Larger and deeper water bodies like lakes Geneva and Bourget are expected to have pronounced effects under higher temperatures in the future in face of their capacity to integrate the effects of meteorological forcing over longer periods of time. Their great heat storage capacities are able to carryover more heat from one year to the next and tend to cool more slowly in autumn and winter (Ficker et al., 2017).

Minor points:

L80: It's unclear how oxygen solubility is a biological indicator. I agree that dissolved oxygen itself is one, but solubility (also as represented here) is merely a function of temperature.

This is a good point. Indeed, the dissolved oxygen concentration is a biological indicator, dependent on the abundance of phytoplankton communities but also on water temperature. Here, we consider oxygen solubility as a proxy for the species' habitats and their capacity to cope with temperature change, an indirect biological indicator (L 83). Indeed, as the oxygen solubility decreases with increasing water temperatures, a decreasing in dissolved oxygen concentrations could be expected and have an impact on fish communities' survival that need a certain amount of oxygen.

L85: I'd change "dissolved oxygen" to "dissolved oxygen solubility"

We agree with this change, which is more accurate.

L97: You're describing all lakes as monomictic but I recall that Lake Geneva did not experience a full turnover for over a decade now. Could you provide references for that statement please?

I thank you for that comment. Indeed, Lake Geneva is generally considered as monomictic but over the past few years, the incomplete mixing of the water column brings it closer to a meromictic lake. I added that precision in the statement (L 102).

L138-142: In most parts of the manuscript, you refer to the scenarios as ssp126, ssp370 and ssp585 only, but here you also refer to them as SSP1, SSP3 and SSP5. Please stick to one naming convention.

I replaced SSP1 and SSP5 by SSP126 and SSP585 to stick to one naming convention, as you recommended (L 150).

L148: Which sensitivity test was carried out and how?

Sensitivity tests were carried out on each climate forcing variables (air temperature, shortwave radiation, wind speed, cloud cover, relative humidity, rain and surface pressure) from MyLake water temperature simulations over a 10-year period, for the 4 perialpine lakes. We added ±20 % to each

Institut National de Recherche pour l'Agriculture, l'Alimentation et l'Environnement (INRAE) - http://www.inrae.fr
CARRTEL Limnology Center, 75 bis, avenue de Corzent – CS 50511, F-74203 Thonon-les-Bains cedex, France

climate variable daily values and then calculated the performance metrics (RMSE and $R^2$) from model outputs and observation data. We calculated the difference between model performance with raw data and after applying the 20% coefficient to the input values. Variables with the greatest variation of its performance metrics were identified as variables with the most important effect on MyLake model performance. All results were summarised in Table S1, in the supplementary.

L152: The mentioning of (ii) and (iv) without (i) and (iii) is slightly confusing. Would it be possible to name these scenarios differently, e.g., (i) airT and SW no correction, (ii) all no correction, (iii) airt and SW with correction, (iv) all with correction?
I have noted the confusion caused by the mentioning (ii) and (iv) without (i) and (iii). As you suggested, I have renamed the 4 configurations in order of appearance in the text (L 168, 169 & 175). I also applied this change in Table 2 to be consistent with the text.

L155: Wouldn't especially Lake Geneva have more meteorological data since the launch of LeXPLORE?
Indeed, meteorological data for Lake Geneva are available on the LeXPLORE data web portal. However, data can be collected only from 2019, which is a shorter period than the 11-years of data available on MeteoSuisse website. Further, meteorological data were only used to reproduce a seasonal pattern and to apply an altitudinal correction factor. Thereby, we considered that a 10-years period of data was long enough to meet these two objectives.

Table 2: Are all fits calculated for all observational data of temperature?
All performance metrics were calculated comparing simulated and all observed data available over a 10- year period, corresponding to the model validation period, specified in Table 5.

L183: I agree with your methodology, and I love how you also compare the fits to long-term time periods, but nonetheless it seems strange that you first mention that L70 "Models are in large extent calibrated against very few years of limnological records so far" but then you are also focusing the calibration/validation on ten years each only. What is the reason for this?
We totally agree with you that the originality of the paper lies to the long-term validation against more than 50 years of limnological records. Nevertheless, we chose to show both 10-years and long-term validation periods for the results to be comparable to the literature in which models are usually calibrated against very few years of limnological records. Furthermore, the 10-years validation period verified the model performance for the four study sites, then the long-term validation validated the robustness of the model for a long-term study.

L207: "[…] epilimnion extent and temperature, hypolimnion extent and temperature, metalimnion upper and lower depths, […]"
This is a good idea to provide clarification on the thermal indices calculated. We added those terms to the text (L 227-228).

L221: "When normal distribution of residuals […]"
Thank you for that suggestion. We added the term "residuals" to be more precise (L 242).

L223: What were the conditions for either choosing Mann-Whitney or Kolmogorov-Smirnov?
The wording may not be very clear in the text, but the difference between the two statistical tests was mentioned in parentheses: Mann-Whitney test was used when normal distribution of residuals was not followed and variances were equal while Kolmogorov-Smirnov test was used when the distribution of residuals was not normal and variances were different.

[Figure]

L252: The thermocline depth fits are quite bad (would they be better if you would only quantify them during stratified summer periods?), but what is the reasoning to quantify them here as percentage of lakes total depth? Although these values sound low, I don't see any reasoning for expressing them this way as an error of 20 m in thermocline depth is still significant and it does not matter if the lake is 300 m deep as the thermocline depth will be in the upper 50 m part anyway.

We appreciate the suggestion raised here. Indeed, when quantifying the performance metrics (especially RMSE) only during the stratified summer period from June to September, the thermocline depth appeared to fit better. We added these new values to Table 7 and modified the text accordingly (L 277-280). We also considered your remark about the irrelevance of estimating the errors as percentage of total depth, as the thermocline depth is generally in the upper 50 m. We have therefore modified RMSE values in the text, and we kept the meter as unit (L 273-277).

L253-260: Why are these fits (DOYs) not included in Table 7?

At first, the reason why these fits were not included was to reduce the size of the table to make it more readable, the complete table being accessible in the supplementary. However, we considered your remark and completed the Table 7 in the manuscript with these fits mentioned in the text.

L345: I recommend deleting "delay" here

As you suggested, we deleted the term "delay" here (L 375).

3.3 Water volumes: habitat: This paragraph is unclear to me. Where these temperature volumes chosen as average temperature output per year, or are these daily data? Also the comparison of non-overlapping distribution areas is quite confusing between two different decades. Could you add more information please?

We apologize for not being clear enough and will add some details into the text to be more precise on the methodology used in here. The water volumes were calculated from daily average water temperature simulated by MyLake model. The bathymetric file allowed us to associate a water volume to each depth. Here, we had a daily average temperature for a certain water volume. Then, we selected only water temperatures exceeding the 3 thresholds (>7°C, >9°C and >12°C) and calculated the sum of these volumes per day. Each day was associated with a certain water volume whose temperature exceeded the 3 characteristic thresholds. In order to compare these water volumes between present (2000-2010) and future (2090-2100), we calculated a daily average over these 10-years periods. At this point, a certain water volume exceeding the 3 thresholds was associated to a day of the year (DOY), for both periods (present and future), according to the 3 scenarios. For the results to be comparable within the 4 lakes, these volumes were reduced as a percentage of the lake total volume. Now, for the 2 different periods, depending on the 3 scenarios, an average lake volume fraction with temperature above 7 °C, 9 °C and 12 °C was calculated for each day of the year. The next step consisted in applying Kernel density to compare both the lake volume fractions with temperature > 7, >9 and >12 °C between the present and future, and also the number of days in the year associated to these lake volume fractions. The non-overlapping distribution showed an increase in both lake volume fractions and number of days associated. We added some methodological details to the text in order to be more specific (L 395-399).

L409: Please keep an eye out for the different digits (. or ,) as here you wrote 12,21 (also the same in the Supplementary tables, please change to dots)

We thank you for that remark. As you suggested, we corrected the number 12.21 (L 441) and replaced all commas by dots in tables in the Supplementary.

Institut National de Recherche pour l'Agriculture, l'Alimentation et l'Environnement (INRAE) - http://www.inrae.fr
CARRTEL Limnology Center, 75 bis, avenue de Corzent – CS 50511,  F-74203 Thonon-les-Bains cedex,  France

[Figure]

L442: Please add references to the ecological thresholds of 10 and 11 mg/L
We thank you for raising that relevant point. These thresholds for dissolved oxygen concentrations (10 and 11 mg/L) were chosen because they correspond to the mean values measured in the epilimnion of the four lakes by OLA (Observatory of Lakes). Thus, we found it interesting to study the evolution of water volume exceeding these thresholds, well adapted to the survival and development of organisms living within the surface layers.

L474: The formulation of delta raise per decade of epi-hypo is very confusing. Could you please state this differently (like just write delta T decade-1)?
We thank you for raising that confusing formulation. As the second reviewer made the same remark, we changed the wording in the text (L 506-517).

L489: Could you please discuss why hypolimnetic temperatures are increasing although water column stability is also increasing? How do these systems still have enough energy for vertical mixing? The same also for ssp126 results in L504: why has that scenario such different results here for epilimnetic and hypolimnetic temperatures?
As you required, we added some explanations to the text about the differences in increase observed in epilimnion and hypolimnion. First, the difference between the upper and deep layers could be linked to a deeper stratification and increase of Schmidt stability. The second argument is relative to the frequency of deep mixing of the water column, which has an effect on the rate of water warming in deep layers (L 517-521).

L519: I'd argue that lateral flow paths and production in littoral zones would also be important for deep and large lake ecosystems.
We thank you for that argument which we added to the text (L 548-549).

L547: I'd argue that Lake Geneva was only the most extensive due to its high depth. Could you check Schmidt Stability trends for the first 50 m of each lake and see if Geneva is still an outlier?
As you suggested, we checked Schmidt stability trends for the first 50 m of each lake to see if Lake Geneva is still outlier. It appeared a small difference from the results observed over the entire water column, as Lake Geneva was still predicted to experience the highest increase of Schmidt Stability according scenarios ssp370 and ssp585, with +2237 and +2947 J m$^{-2}$ decade$^{-1}$, respectively. A smaller difference was observed but still remained. Yet, same trends were observed, as you can see on the Figure S5 in the supplementary.
We also added a table (Table S7 in the supplementary) showing every result for both Schmidt stability calculated on the entire water column or only on the first 50 m.

L591: Please add information to which approach you are referring here, I assume it's the focus on air temperature and short-wave radiation, right?
Indeed, this is the approach developed in that study we were referring to in that statement. As you suggested, we specified more details in the text to be clearer (L 600-601).

L645: The sentence about "Differences in the oxygen solubility response to climate change are also observed" seems very vague to me.
We agree that this sentence was not very clear. Thus, we chose to remove it from the text, as it did not provide any further clarifications and was redundant to the previous paragraph.

Institut National de Recherche pour l'Agriculture, l'Alimentation et l'Environnement (INRAE) - http://www.inrae.fr
CARRTEL Limnology Center, 75 bis, avenue de Corzent – CS 50511, F-74203 Thonon-les-Bains cedex, France

[Figure]

Reviewer #2:

This paper presents result of model projections of the temperature and mixing under past and future climates (scenarios ssp125, 370 and 585) for four peri-alpine lakes, using the IPSL-CM6-LR atmospheric model from ISIMIP3b with the 1D lake model MyLake. It also considers future changes in thermal habitat and oxygen solubility of these lakes, making inferences about their suitability as a habitat for some fish species. Good projections, eg as used with the new ISIMIP3b data, of the potential effects of warming on these important lakes will be very valuable and an important and novel research result.

My main issue with the paper concerns the style. I found it quite confusing and the rationale behind the study in the introduction as well as the methods should be more clearly stated. For instance, I did not properly understand why the authors chose to use only shortwave radiation and air temperature projections for model hindcasts and future projections, and use historical means for the other model forcing variables. Also it was not so easy to quickly understand how this was done. I think you show show the comparison between the model configurations with just shortwave and airtemp, and with all forcings from the GCM more clearly and make a short justification for your chosen configuration. I also didn't understand why the authors began with an ensemble of climate and lake models, but selected only one of each for the analysis, or what the value is of performing a long hindcast. This all might sound provocative, and I am not saying this is unjustified at all, only that it was not explained sufficiently to the reader. The scientific results seem sound, but I struggled to interpret the plots (which have a resolution and font size that are too low for me to read the axes), and I think the discussion needs to be reworked, focusing on the interpretation rather than the description of the results. There are some interpretations to explain the differences between the lakes, but I think they need to go deeper, like if simply wind fetch is really causing deeper mixing in Geneva, or the effect of ealier stratification onset timing for deep water warming trends. Several formulations in the manuscript describe complex things, but need to be sharpened to make this complexity more understandable. In the sections below I have tried to point out clearly the parts that I am referring to.

We are grateful to Reviewer #2 for your careful revision of our manuscript, making constructive and valuable suggestions. The discussion was revised and new content was implemented to provide further interpretations on the results.

***Abstract***
General: in this modelling context, try to use "parameters" only to refer to model parameters, and use "variables" for things like wind (eg line 62)or e.g. "substances" when talking about concentrations (eg line 20).

We thank you for that precision. As you suggested, we replaced the term "parameters" by "variables" to refer to forcing variables used as inputs (L 23 and L 29), and "substances" when talking about concentrations (L 21).

***Introduction***
You go into some detail to describe the approach, but not quite enough to make it clear, mainly in lines 75-86. Later in the methods you go into a little more detail, but I found it unclear when starting with the introduction whether you use observed radiation and airtemp to force the model, and then estimated the other variables? It would be helpful to describe the rationale behind this. The introduction should be focused more to justify and define the main research question. For instance, there is no mention of the thermal habitat and I don't recall that you reported or discussed any results based on the 4 different model configurations.

Institut National de Recherche pour l'Agriculture, l'Alimentation et l'Environnement (INRAE) - http://www.inrae.fr
CARRTEL Limnology Center, 75 bis, avenue de Corzent – CS 50511,  F-74203 Thonon-les-Bains cedex,  France

[Figure]

Line 129-130: did you use the historical scenario for each model (I assume)? Please state explicitly. The historical scenario from the IPSL model is available back to 1850 for all required forcing variables for the lake model. As mentioned above, please explain why you use only shortwave and airtemp.

We thank you for that remark. As you suggested, we stated explicitly in the text the scenarios from which data were collected (L 142-144). Both historical and ssp370 (intermediate) scenarios were used to cover the whole period of observation data (1987-2019). Indeed, all required forcing variables were available back to 1850 from IPSL model. We chose to use only shortwave and air temperature to keep only variables with the highest level of confidence over the long term, as the long-term assessment of global warming on the peri-alpine lakes thermal regime was both the particularity and objective if that study. Thus, we added a sentence to the text to explain more precisely the reason why we used only those two variables to force the model (137-139).

Line 132-133: sentence not clear (350 data)
We rephrased that sentence to be more precise. (L 144-145)

Line 148-150: It is unclear what is meant by the sensitivity test – could you please briefly elucidate, even if the description is in the supplement? This hypothesis needs to be more clearly stated in the introduction

Same question was raised by the first reviewer. You will find below our answer with details on the sensitivity test carried out on the forcing climatic variables:

"Sensitivity tests were carried out on each climate forcing variables (air temperature, shortwave radiation, wind speed, cloud cover, relative humidity, rain and surface pressure) from MyLake water temperature simulations over a 10-year period, for the 4 perialpine lakes. We added ±20 % to each climate variable daily values and then calculated the performance metrics (RMSE and $R^2$) from model outputs and observation data.  We calculated the difference between model performance with raw data and after applying the 20% coefficient to the input values. Variables with the greatest variation of its performance metrics were identified as variables with the most important effect on MyLake model performance. All results were summarised in Table S1, in the supplementary."

As you suggested, we stated it more clearly in the introduction (L 160-166).

Line 151-160: it would be simpler if you described the configurations in order i), ii), iii), iv), rather than beginning with ii and iv.

The same suggestion was made by the first reviewer. We have modified the order of the 4 configurations accordingly (L 168, 169, 175 and Table 2).

Table 2: I don't completely understand the table from the legend. Do the performance metrics represent the lake model temperature errors (units degrees Celsius)? The configurations are not really understandable in this context – when you say only air temperature and shortwave, do you mean that these variables come from the climate model IPSL-CM6-LR, while the other input variables are observed data?

Indeed, the performance metrics have been calculated between simulated and observed water temperature, in Celsius degrees. We added the unit in Table 2 in order to be clearer. We also added explanations of the 4 tested configurations in the Table 2 legend, in order to be more understandable. The different configurations were also detailed in the text (L 184-187). Indeed, in the configurations (i) and (iii), only air temperature and shortwave radiation were extracted from ISIMIP3B – IPSL, whereas other variables were calculated from meteorological observations. For these variables, daily means were calculated and then replicated every year to apply a seasonal pattern.

Institut National de Recherche pour l'Agriculture, l'Alimentation et l'Environnement (INRAE) - http://www.inrae.fr
CARRTEL Limnology Center, 75 bis, avenue de Corzent – CS 50511,  F-74203 Thonon-les-Bains cedex,  France

[Figure]

Line 181-184: I didn't understand this bit – you should add more specific details (eg what is the instrumental period and why do you need to identify similar climate conditions, what do you mean by the RMSE between winter air temperatures, etc?

We have considered your remark raised here. Accordingly, we added some details into the text to clarify the method used to estimate the water temperature initial profile (L 210-215). The objective was to identify the year with available water temperature observation data (from OLA database), with the closest climatic conditions to the year 1850, for which no field data were available. Air temperature from January 1st to January 31st was compared between data extracted from ISIMIP3b – ISPSL and meteorological data over the period corresponding to OLA database. We calculated RMSE between these years, and identified the year of meteorological data with the closest air temperature to January in 1850. Water temperature profile was then collected for that specific year and used as initial profile in January 31st 1850.

Line 201-202: Be more precise about the formulation and units – Ts at 5 m (not Ts=5m), and add units for Tb (Tb at 60 m, 60m, 299 m, and 140 m depth for …)

We thank you for having noted this inaccuracy and lack of units. We have modified the text accordingly (L 221-222).

How did you define stratification duration – is it the total number of stratified days per year (where each period > 5 days), or the duration of the longest uninterrupted period?

The stratification duration was defined as the longest uninterrupted period of stratification. Indeed, it was calculated from the number of days between onset and end-of-stratification, themselves defined as the day when Ts-Tb >2°C during more than 5 consecutive days. We added details about this method in the text to address the inaccuracy (L 230-231).

**Results**

Note: the resolution of the figures is very low and the labels are very small, so that I am unable to read the axis labels on any of them except Fig 1. Please bear this in mind if there are potential interpretation differences.

We thank you for raising the issue of the resolution of the figures. All axis labels have been enlarged and the quality of the figures improved.

Line277-281: rather report change/decade only for trends over time. If you make a comparison of period means (eg mean 1990-2020 with mean 2070-2100), then report the change as absolute, eg an increase of 1.5 degrees Celsius from the baseline period to the end of the century.

As you suggested, we added the rates of increase between the two periods as absolute to better represent the expected air temperature and shortwave radiations variations (L 305-310).

Figure 4: it looks like you applied some smoothing to the time series – please describe what you did. Also, I suggest to label the panels only with letters, not combined letters and numbers

On the Figure 4, a locally weighted smoothing was applied to air temperature and downwelling shortwave radiation time series over the period 1850-2100, in order to better identify the trends. We added that precision in the figure title to specify the method used (L 317). As for the labelling, we agree that using only letters would add homogeneity within the panels and would be maybe more understandable. However, we feel like using both letters and numbers to refer to the two variables on one hand (air temperature and shortwave radiation) and to the three scenarios on the other hand, is easier to describe in the legend. Still, we would agree to make some modifications if you think this is necessary.

[Figure]

Line 297: I am curious about the negative temperature trends you report here, and also interpret in the discussion. I cannot see any trend in Fig 5 – the epilimnion temperature always looks to be increasing in ssp126.

We thank you for raising this question. Cooling surface trends in the future were previously predicted in large deep lakes in Central Europe (Schmid and Köster, 2016). The mechanistic effect of surface cooling in the most optimistic scenario over 1970–2100 is explained by the future development of both air temperature and solar radiation predicted to decrease. Higher solar radiation trends observed over entire Central Europe in the last decades largely caused by decreasing aerosol contents in the atmosphere as a consequence of air quality measures are not expected to continue to a similar extent in the future, thereby the recent past trend of surface warming is not expected to persist in the future. This discussion was implemented on L 510-515. In Figure 5, the negative trends at -0.07 to -0.08 °C decade–1 are indeed difficult to visualize in the averaged curves; they can be better visualized by the annual dots.

Line 316-318 (and corresponding methods section): the non-overlap in the temperature density distributions is not sufficiently described to understand what it represents. Which temperatures are being compared exactly? They are obviously averaged, but how (eg across the water column, but then they may need to be volume weighted)?

Same question was raised by the first reviewer. You will find below the answer, with precisions on the non-overlap in the temperature density distributions:

"The water volumes were calculated from daily average water temperature simulated by MyLake model. The bathymetric file allowed us to associate a water volume to each depth. Here, we had a daily average temperature for a certain water volume. Then, we selected only water temperatures exceeding the 3 thresholds (>7°C, >9°C and >12°C) and calculated the sum of these volumes per day. Each day was associated with a certain water volume whose temperature exceeded the 3 characteristic thresholds. In order to compare these water volumes between present (2000-2010) and future (2090-2100), we calculated a daily average over these 10-years periods. At this point, a certain water volume exceeding the 3 thresholds was associated to a day of the year (DOY), for both periods (present and future), according to the 3 scenarios. For the results to be comparable within the 4 lakes, these volumes were reduced as a percentage of the lake total volume. Now, for the 2 different periods, depending on the 3 scenarios, an average lake volume fraction with temperature above 7 °C, 9 °C and 12 °C was calculated for each day of the year. The next step consisted in applying Kernel density to compare both the lake volume fractions with temperature > 7, >9 and >12 °C between the present and future, and also the number of days in the year associated to these lake volume fractions. The non-overlapping distribution showed an increase in both lake volume fractions and number of days associated. We added some methodological details to the text in order to be more specific (L 395-399)."

Line339-351: I think it would make more sense to analyse the trend over the whole period, or if you compare the baseline 1990-2020 with the future 2070-2100, then it would be more informative to report the absolute change rather than the trends today and in the future. This would put into perspective the fact that ssp126 initially warms but then cools slightly towards 2100 reflecting emissions reductions. So although there is a cooling trend at the end of the century in this scenario, this should not be confused with the overall long-term warming trend compared to today. Also, in Table S4, MyLake has an R2 of practically zero for stratification phenology in many instances – if you trust there results, you should mention the reliability in the discussion. Also, I wonder if these results would be better in a table, which makes comparison and overview of the changes much easier.

[Figure]

We thank you for this very good remark that we have taken into consideration by integrating absolute changes between the two periods (374-378). About the very low R² for stratification phenology, this can be explained by the frequency of the lakes monitoring (twice a month), which does not allow us to capture the exact dates of onset and end-of-stratification. We would require the complete time series data to be able to compare to the dates predicted by the model. As other thermal indices, such as Schmidt stability, were predicted with good precision by the model, we chose to trust variables related to stratification phenology. A paragraph was added to the discussion to clarify this choice (L 625-629).

Figure 7: I suggest to reverse the colours – blue for the coolest scenario and red for the warmest.
We thank you for that very good point. As you suggested, we have inverted the colours, with blue for the coolest scenario and red for the warmest. We made this change for both Figure 7 and Figure 9.

Line 368: do you mean that temperature exceeded 7 degrees in all scenarios in the future period? Are these daily temperatures or a mean (Methods say averaged daily data, but this is not quite clear)? The background to my question is that it is quite different if the lake never cools below 7 degrees, even in winter, or if there are only a few days in summer when this threshold is exceeded.
Indeed, it appeared that in all 3 scenarios in the future period (ssp126, ssp370 and ssp585 from 2090 to 2100), the entire water column will reach 7 °C. First, water volumes and lake volume fractions with temperature above 7 °C were calculated at daily time step. Then, the annual mean of these water volumes were calculated. The results showed that annual average of lake volume fractions reached 100 % of the lake total volume, during the whole year, including winter period. We added some explanations in the text to be clearer (L 395-399). We also added new figures in the supplementary to illustrate the evolution of the lake volume fractions with temperature above 7 °C, for Lake Bourget (Figure S6) and Geneva (Figure S7) according to ssp126 scenario. We chose the most optimist scenario to show that even in that case, the whole entire water column will exceed 7°C throughout the year, at the horizon 2100.

Line 371:  Better say 100% non-overlap rather than 100% increase.
As you suggested, we changed the wording "increase" to "non-overlap" to be more specific (L 403).

Lines 440 ff: Section on oxygen solubility density overlap: I feel that it does not make sense to report the non-overlap of the density distribution of oxygen solubility. This is hard for me to meaningfully interpret, and I suggest to remove this part.
We understand the difficulty to interpret this part. As a result, we added further details about the Kernel density method tried to explain more precisely the way of interpretation (L 475-478). If that part remained meaningless to you, we could remove it from the text.

***Discussion***

The discussion section contains a large repetition of the results, and there should be more integration of the findings into the context of the literature, with the aim of detecting some of the important mechanisms driving the thermal changes. You may like to consult Kraemer et al (2021) and literature therein to interpret lake thermal habitat (Nature Climate Change 11, 521-529). Plus there is a lot of literature on the thermal properties of these peri-alpine lakes eg Geneva that may help identify the mechanisms why these lakes behave differently. Altogether, I think the discussion and conclusion should be heavily revised and especially remove the results descriptions.

[Figure]

The discussion section was revised accordingly on L 506-664. We are grateful for the literature indication which brought contributions to our discussion. The new citation was included on the Reference list.

Line 469: check year
We thank you for noticing this typo. We modified that error in the text (L 510).

Line 473-475: you can remove this repetition of results and just concentrate on the interpretation. If you want to mention this, it would be simpler to say that surface water is warming faster than deep water at 0.07 degrees/decade in Lake Bourget and at 0.15-0.16 degrees/decade in the other lakes.
As you suggested, we changed the wording of that sentence to make it easier to understand (L 499-526).

Line 481-485: as I mentioned above, you should be careful with reporting ssp126 in this way. Although ssp126 shows cooling air temperatures at the end of the century (negative trends due to emissions reductions), it still represents a warmer future, so make sure your formulations are not misleading. I find the formulation as it is now sounds like the surface is cooling compared to today. Also, I could not see the negative trends you are talking about in Fig 5, perhaps you could comment on this?
We agree that this sentence needs clarification. This discussion was accordingly revised on L 510-515 to provide the reasoning behind the cooling trends in the future. In Figure 5, the negative trends at -0.07 to -0.08 °C decade$^{-1}$ are indeed difficult to visualize in the averaged curves; they can be better visualized by the annual dots.

Lines 489-497: this is all just a repetition of the results and is unnecessary.
Agreed. The discussion section was revised accordingly on L 519-521 to avoid repetition of the results and to provide more integration of the findings into the context of the literature.

Line 511 ff: You shouldn't call this section Oxygen Trends because you didn't model oxygen, just temperature, on which the oxygen solubility is only calculated. This section is essentially a repetition of the results. You should focus on interpreting the results, explaining their significance, drivers, and context.
Agreed. The Oxygen trends section was revised accordingly on L 530-554 to avoid repetition of the results and to focus on interpreting the results.

Lines 542-551: You need to account for the differences in lake morphology when comparing Schmidt stability. Lake Geneva will need much more energy to mix it than Lake Annecy. Perhaps comparing relative changes would be more meaningful than absolute changes.
We totally agree with your interpretation and the need to account for the differences in lake morphology when comparing Schmidt Stability. Accordingly, we added some explanations and comparisons in the text (L 558-562).

Institut National de Recherche pour l'Agriculture, l'Alimentation et l'Environnement (INRAE) - http://www.inrae.fr
CARRTEL Limnology Center, 75 bis, avenue de Corzent – CS 50511, F-74203 Thonon-les-Bains cedex, France

---

## Author Response (AR2)

Le Bourget-du-Lac, January 17th 2023

Dear Hydrology and Earth System Sciences Editors,

We would like to thank the *Hydrology and Earth System Sciences* Editors for giving us one more opportunity to revise the manuscript, and the editor for its constructive comments on our manuscript « *Past and future climate change effects on thermal regime and oxygen solubility of four peri-alpines lakes* » by O. Desgué-Itier, L. Soares, O. Anneville, D. Bouffard, V. Chanudet, P-A. Danis, I. Domaizon, J. Guillard, T. Mazure, N. Sharaf, F. Soulignac, V. Tran-Khac, B. Vinçon-Leite, J-P. Jenny. Please find enclosed a revised manuscript that takes into account the concerns raised by the editor.

We carefully considered new comments, and accordingly, we added statements to clarify why only MyLake model was used instead of the other models. As recommended, the abstract has been revised to better highlight research questions and the value of this study. Besides, we did a careful proofreading to correct grammatical and typing errors.

Please find below in blue text our responses to the comments.

Sincerely yours

Olivia Desgué-Itier
"Ingénieur d'Etudes"
Now at CARRTEL Limnology Center, France
olivia.desgue@inrae.fr

Institut National de Recherche pour l'Agriculture, l'Alimentation et l'Environnement (INRAE) - http://www.inrae.fr
CARRTEL Limnology Center, 75 bis, avenue de Corzent – CS 50511,  F-74203 Thonon-les-Bains cedex,  France

[Figure]

Response to the editor' s general comments:

\* More generally, the first half of the abstract is relying on a few introduction type sentences, rather than specifying a clear aim. I would ask that you try to make the opening more concise and make clear the research questions (rather than just saying 1D models have shortfalls). The interesting question of this study is buried in a later methodological statement: "the effects of climate change on the thermal regime and oxygen solubility were analyzed …" . But why are they analysed? What are the questions the study answers?

We thank you for pointing this out. We have modified the abstract accordingly, more specifically the first part, to clarify the research questions and to insist on the interests of that study.

\* Regarding the model inter-comparison – this is given some weight in the abstract ("MyLake is the best …"), but in the main paper this comparison is in supplementary material. There has been some discussion of this in the review and I'm satisfied, except I do have the position that it is not 1 model that is the "best" but the implementation of one model that was the best … in this case. I say this as the parameter vector optimised for each model seems limited and for example there are parameters and model options that were excluded (e.g., deep mixing in GLM) and so it's hard from this assessment to tell if one model is truly better than other. As developer of GLM I am somewhat biased so I will leave it to the authors to decide how to navigate this, but my suggestion is to simply qualify the language slightly by saying that "of the models implemented for this analysis, MyLake performed…. ".

We have taken into account your interesting remark and modified the text accordingly. Indeed, we explained more precisely that the models' performances were dependent on the implementation and the parameters tested for each model.

\* Last line: "These results suggest important degradation in lake thermal and oxygen conditions and a loss of habitats for endemic species." Strictly, the paper does not prove their will be a loss of habitat for a specific species and this is inferred by a change in temp, and in solubility from ~10.75 down to 9 mg/L. The discussion on line 575 talks generally about sensitivity to temperature and this makes the case that some species have physiological cues that may be impacted upon. The paper however does not answer how important the oxygen concentration change is from a habitat point of view so the wording of the abstract is therefore potentially misleading in this regard. I'm not sure from reading the paper (section 4.2) if there are species that are so sensitive to a change in oxygen concentration? Noting that this will still be 100% saturation, which I understand from fish ecophysiologists is more important than the absolute oxygen concentration. Usually species come under pressure at lower oxygen saturation values like <50%. Therefore, whilst I agree you need a significance statement in the abstract along these lines, more precise use of language should be used for this, and I would also suggest that further discussion (with references) about implications of change in oxygen saturation on fish habitat is warranted. For example, see also Magee et al 2018 (CJFAS Vol 75 "Modeling oxythermal stress for cool-water fishes in lakes using a cumulative dosage approach"

As you suggested, we modified the last line of the abstract to be more precise about the potential impact of changes in temperature and oxygen conditions. We no longer refer to a loss of habitats but we are referring to changes in habitat conditions (L36-38). We also added some references in the discussion (L585-586 & L594-596).

Institut National de Recherche pour l'Agriculture, l'Alimentation et l'Environnement (INRAE) - http://www.inrae.fr
CARRTEL Limnology Center, 75 bis, avenue de Corzent – CS 50511, F-74203 Thonon-les-Bains cedex, France

[Figure]

Response to specific comments:

* Line 20 lake modelS (plural)
* Line 21 such AN approach (missing word)
* Line 23: Please reword to be more concise – I think there is some discussion about downscaling and not sure what is meant by "serious" - " ... which have several limitations that are barely discussed, such as the need of serious downscaling "
* Line 24 – extenD, not extenT

Following suggestions, the term "serious" has been replaced by "significant" (L25); the identified spelling mistakes were corrected.
Comments made in the PDF have been addressed and are highlighted in blue in the manuscript. The line numbering refers to the new version of the manuscript (c.f. Draft_HESS_version_RC1_RC2_last_version_blue.docx).

In the introduction:
* Line 53 (and line 70): "the study of lakes' thermal regime over decadal to centennial timescales is still limited" . There are modelling papers that validate over long-time periods, (eg: Magee & Wu, HESS, 21, 6253–6274, 2017; Magee and Wu, Hydrol Proc 31 308-323, 2017; Ayala et al 2020 already cited in discussion but relevant here), so I think these statements are overly general and could be more precise.

We thank you for that very good point. We added some details about the limits of existing studies, considering the literature you advised us to refer to (L54).

* Line 65 : papers like Sadeghian et al (Hydrol Sci J, Vol 67, 2022) have compared lake model sensitivity to different weather forcing products
Tkank you for that reference which has been added to the introduction (L66-67).

* In the discussion, I was wondering if Section 4.2 sits better after section 4.3? Maybe I misunderstand the flow of logic but, but its seems 4.1 and 4.3 are about past and future temp (respectively), and 4.2 is about oxygen, so I would suggest the segue to oxygen after discussion of temp may suit better? Not a requirement of the revision but just for the authors to consider, also in light with my comment about oxythermal habitat discussion made above.

We thank you for that very good point. The idea was to first discuss the evolution of the two studied variables and then to interpret and compare the sensitivity of each lake regarding these variables. But as you noticed, it appeared more logical to move 4.3 before 4.2 as it focuses on temperature, as then to discuss about the oxygen conditions.

Overall, I still note quite a few grammatical and small editorial issues (eg missing space here and there, extra letter etc), so request you do a thorough proofread to check for these issues.
We did a complete review of the article to correct the grammatical and typing errors.

Institut National de Recherche pour l'Agriculture, l'Alimentation et l'Environnement (INRAE) - http://www.inrae.fr
CARRTEL Limnology Center, 75 bis, avenue de Corzent – CS 50511,  F-74203 Thonon-les-Bains cedex,  France

---

## Author Response (AR3)

Le Bourget-du-Lac, February 1st 2023

Dear Hydrology and Earth System Sciences Editors,

We would like to thank the *Hydrology and Earth System Sciences* Editors for their comments on our manuscript « *Past and future climate change effects on thermal regime and oxygen solubility of four peri-alpines lakes* » by O. Desgué-Itier, L. Soares, O. Anneville, D. Bouffard, V. Chanudet, P-A. Danis, I. Domaizon, J. Guillard, T. Mazure, N. Sharaf, F. Soulignac, V. Tran-Khac, B. Vinçon-Leite, J-P. Jenny. Please find enclosed a revised manuscript that takes into account the concerns raised by the editor.

In the revised version we corrected the Bruce et al. 2018 citation in the text and references. We also added the missing final point at the end of the abstract.

Sincerely yours

Olivia Desgué-Itier
"Ingénieur d'Etudes"
Now at CARRTEL Limnology Center, France
olivia.desgue@inrae.fr

Institut National de Recherche pour l'Agriculture, l'Alimentation et l'Environnement (INRAE) - http://www.inrae.fr
CARRTEL Limnology Center, 75 bis, avenue de Corzent – CS 50511,  F-74203 Thonon-les-Bains cedex,  France